

# A Python interface to the Fortran-based Parallel Data Assimilation Framework: pyPDAF v1.0.0

Yumeng Chen[1,2], Lars Nerger[3], and Amos S. Lawless[1,2]

[1]School of Mathematical, Physical and Computational Sciences, University of Reading, Reading RG6 6ET, UK
[2]National Centre for Earth Observation, University of Reading, Reading RG6 6ET, UK
[3]Alfred-Wegener-Institut, Helmholtz-Zentrum für Polar-und Meeresforschung (AWI), 27570 Bremerhaven, Germany

**Correspondence:** Yumeng Chen (yumeng.chen@reading.ac.uk)

**Abstract**

Data assimilation (DA) is an essential component of numerical weather and climate prediction. Efficient implementation of DA benefits both operational prediction and research. Currently, a variety of DA software programs are available. One of the notable DA libraries is the Parallel Data Assimilation Framework (PDAF) designed for ensemble data assimilation. The DA framework is widely used with complex high-dimensional climate models and is applied for research on atmosphere, ocean, sea ice and marine ecosystem modelling, as well as operational ocean forecasting. Meanwhile, there exists increasing need for flexible and efficient DA implementations using Python due to the increasing amount of intermediate complexity models as well as machine learning based models coded in Python. To accommodate for such needs, here, we introduce a Python interface to PDAF, pyPDAF. The Python interface allows for flexible DA system development while retaining the efficient implementation of the core DA algorithms in the Fortran-based PDAF. The ideal use-case of pyPDAF is a DA system where the model integration is independent from the DA program, which reads the model forecast ensemble, produces a model analysis and update the restart files of the model, or a DA system where the model can be used in Python. With implementations of both PDAF and pyPDAF, this study demonstrates the use of pyPDAF and PDAF for coupled data assimilation (CDA) in a coupled atmosphere and ocean model, the Modular Arbitrary-Order Ocean-Atmosphere Model (MAOOAM). Using both weakly and strongly CDA, we demonstrate that pyPDAF allows for the utilisation of Python-based user-supplied functions in the Fortran-based DA framework. We also show that the Python-based user-supplied routine can be a main reason for the slow-down of the DA system based on pyPDAF. Our CDA experiments confirm the benefit of strongly coupled data assimilation compared to the weakly coupled data assimilation. We also demonstrate that the CDA not only improves the instantaneous analysis but also the long-term trend of the coupled dynamical system.

## 1 Introduction

Data assimilation (DA) is widely used in weather and climate modelling where observations are used to constrain the model prediction based on the uncertainty of both the observations and the model forecast. Due to the limited predictability and





imperfect models, DA has become one of the most important techniques for the numerical weather and climate predictions. Progresses of the DA methodology development can be found in various review articles and books (e.g., Bannister, 2017a; Carrassi et al., 2018; Vetra-Carvalho et al., 2018; Evensen et al., 2022).

To ameliorate the difficulties in the implementation of different DA approaches, several DA software programs and libraries have been proposed (e.g., Nerger et al., 2005; Anderson et al., 2009; Raanes et al., 2024; Trémolet and Auligne, 2020). Even though the implementation of the core DA algorithms is similar, these software programs/libraries are typically tailored to different purposes. For example, the Joint Effort for Data assimilation Integration (JEDI) is a piece of self-contained software that includes a variety of functionalities that can be used for all aspects of a DA system mainly for operational purposes while DA software for methodology research such as DAPPER is designed for identical twin experiments equipped with low complexity models.

One widely used DA framework is the Parallel Data Assimilation Framework (PDAF) developed and maintained by the Alfred Wegener Institute (Nerger et al., 2005; Nerger and Hiller, 2013). The framework is designed for efficient implementations of ensemble-based DA systems for complex weather and climate models. The DA implementations require user-supplied functions to provide case-specific information about the DA system including the treatment of observations and localisation. A variety of successful use-cases of PDAF were developed for complex weather and climate models. For example, an ensemble DA system was developed for the Alfred-Wegener-Institute Climate Model (AWI-CM, Sidorenko et al., 2015) using PDAF (Nerger et al., 2020). The framework is also used with the Los Alamos Sea Ice Model (CICE) to develop an Arctic sea ice DA system to assimilate CryoSat-2 sea ice thickness datasets (Williams et al., 2023). In the case of land surface modelling, PDAF is coupled with the Community Land Model version 5 (CLM5) by Strebel et al. (2022). Further use-cases of PDAF can be found in the PDAF website (PDAF, https://pdaf.awi.de). Even though PDAF provides a highly flexible framework for the DA system, the implementation of user-supplied functions still require additional code development, which can be time-consuming especially when the routines have to be written in Fortran, a popular programming language for weather and climate applications.

In recent years, Python is gaining popularity in weather and climate communities due to its flexibility and ease of implementation. For example, Python is adopted by some low- to intermediate-complexity models (e.g., De Cruz et al., 2016; Abernathey et al., 2022), models with a Python wrapper (e.g., McGibbon et al., 2021), and machine learning based models (e.g., Pathak et al., 2022; Lam et al., 2023; Bi et al., 2023). For the application of DA, DAPPER provides a variety of DA algorithms for twin experiments using low-dimensional Python models. The Ensemble and Assimilation Tool, EAT (Bruggeman et al., 2023) was proposed to set up a 1D ocean-biogeochemical DA system. The Python tool only has a Python interface to a few PDAF routines while the rest of the system is coded in Fortran including the 1D ocean-biogeochemical model, GOTM-FABM.

Here, we introduce a Python interface to PDAF, pyPDAF. Compared to the user-supplied functions implemented in Fortran, the Python-based implementation can facilitate code development thanks to a variety of packages readily available in Python. In the meantime, DA algorithms that are efficiently implemented in Fortran can still be utilised. Using pyPDAF, one can implement a Python-based offline DA system where the model output is written onto a disk. If a numerical model is available





in Python, pyPDAF allows for online DA system implementation where DA algorithms can be used with in-memory data exchange without I/O operations.

In this study, we demonstrate the use of pyPDAF in a coupled data assimilation (CDA) setup. The research on CDA is motivated by the use of coupled earth system models, especially for the coupled atmosphere and ocean simulations (Eyring et al., 2016; Walters et al., 2019). Traditionally, each model component is assimilated individually and the state of each model component interacts with the others only in the coupled model forecast. This approach is called weakly coupled DA (WCDA). It is desirable to perform DA jointly for all model components simultaneously, usually denoted as strongly coupled DA (SCDA). Studies report a suite of benefits of using SCDA. For example, Smith et al. (2015) shows that the SCDA can improve dynamically balanced analysis leading to reduced initialisation shocks. Sluka et al. (2016) reported improvements in analysis with SCDA in an intermediate complexity model. Tang et al. (2021) performed SCDA of ocean observations into the coupled atmosphere-ocean model AWI-CM and found positive effects in particular in the polar regions. Further studies can be found in a suite of review articles on CDA (Penny and Hamill, 2017; Zhang et al., 2020; de Rosnay et al., 2022; Kalnay et al., 2023).

In this paper, we will first introduce ensemble-based data assimilation, the principal objective of PDAF, in Sect. 2. Section 3 will describe the design and implementation of PDAF and pyPDAF. In Sect. 4, the concept of CDA will be discussed. In Sect. 5, the experiment and model setup will be described. Section 6 will report the performance of PDAF and pyPDAF in CDA setup. We will conclude in Sect. 7.

## 2 Ensemble-based data assimilation

The parallel data assimilation framework focuses on ensemble-based DA methods. Ensemble-based DA is a class of DA approaches that approximate the statistics of the model state and its uncertainty using an ensemble of model realisations. The ensemble-based DA was motivated by DA approaches based on Bayes theorem where the prior, typically a model forecast, and posterior (analysis) distributions can be approximated by a Monte Carlo approach. This allows for an embarrassingly parallel implementation which means that, with sufficient computational resources, the wall clock computational time does not increase with the ensemble size.

Under the Gaussian assumption of the forecast and analysis distributions, one of the most notable ensemble-based DA methods is the ensemble Kalman filter, EnKF (Evensen, 1994). The EnKF approximates the forecast and analysis error distribution by an ensemble under the Gaussian assumption. The method was proven to be successful in many applications (e.g., Houtekamer et al., 2005; Feng et al., 2009; Hamill et al., 2011; Sakov et al., 2012). To further improve the efficiency and reliability of the EnKF, multiple variants of the EnKF were proposed, such as singular evolutive intepolated Kalman filter (SEIK, Pham, 2001), ensemble transform Kalman filter (ETKF, Bishop et al., 2001), error space transform Kalman filter (ESTKF, Nerger et al., 2012), and the deterministic ensemble Kalman filter (Sakov and Oke, 2008). In practice computational resources limit the feasible ensemble size in the high-dimensional realistic DA applications in the Earth system. The ensemble-based DA approaches typically suffer from sampling errors from limited ensemble size. To counter these deficiencies, covariance matrix





inflation and localisation are commonly used (e.g., Pham et al., 1998; Hamill et al., 2001; Hunt et al., 2007). In particular, the domain localisation is tailored for efficient parallel implementations that are commonly used in high-dimensional DA systems.

Ensemble-based DA can also treat fully non-linear non-Gaussian problems. The most notable example is particle filters (van Leeuwen et al., 2019). They can be used to solve fully non-linear problems without assumptions on the prior and posterior distribution. However, for high-dimensional geoscience applications, the classical particle filters suffer from the "curse of dimensionality" where the required ensemble size grows exponentially with the dimension of the state vector making the approach computationally infeasible. Recent developments of the particle filters significantly improve the stability and reduce the required ensemble size of the approach making it a potential choice for low-to-medium complexity models, such as implicit equal-weights particle filters (Zhu et al., 2016) and the particle flow filter (Hu and van Leeuwen, 2021). An overview of other developments of particle filters can be found in van Leeuwen et al. (2019).

The ensemble-based DA approaches are adopted by many operational centres where traditionally variational methods are used (e.g., Clayton et al., 2013; Caron et al., 2015; Bonavita et al., 2016; Hersbach et al., 2020). In variational methods, ensemble approaches are used to achieve flow-dependent background covariance matrix, and/or to avoid explicit computation of the adjoint model in the minimisation process by using an ensemble approximation. These goals can be realised using various different methodologies and a detailed review of these methods can be found in Bannister (2017b).

## 3 PDAF and PyPDAF

The Parallel Data Assimilation Framework (PDAF) is designed for research and operational DA systems. As a Python interface to PDAF, pyPDAF uses the DA algorithms implemented in PDAF and the same implementation approach to build a DA system.

### 3.1 Parallel Data Assimilation Framework (PDAF)

The parallel data assimilation framework, PDAF, is a Fortran-based DA framework providing fully optimised, parallelised ensemble-based DA algorithms. The framework provides a software library and defines a suite of workflows based on different DA algorithms provided by PDAF including various ensemble Kalman filters/smoothers, ensemble-based 3DVar, particle filters and other non-linear filters. To deal with sampling errors in the ensemble-based DA, the framework also provides options for adaptive inflation schemes and for localization.

As a framework for ensemble DA, the framework comes with the functionality to generate the initial ensemble. The ensemble is generated based on the model trajectory of the modelled truth using the second-order exact sampling (Pham, 2001). The assumption is that the uncertainty of the model initial condition lies in the phase space of the model trajectory. The space is represented by the singular values and its corresponding vectors using an empirical orthogonal function (EOF) decomposition. In the second-order exact sampling an ensemble of $N_x$-dimensional state vectors and $N_e$ ensemble members is generated by perturbations that preserve the mean and represent, up to the rank $N_e - 1$, the same covariance matrix as the singular vectors. This is achieved by multiplying a matrix, $\mathbf{A} \in \mathbb{R}^{N_x \times (N_e-1)}$, consisting of $N_e - 1$ singular vectors scaled by the singular values with an orthonormal random matrix generated from Householder reflections.



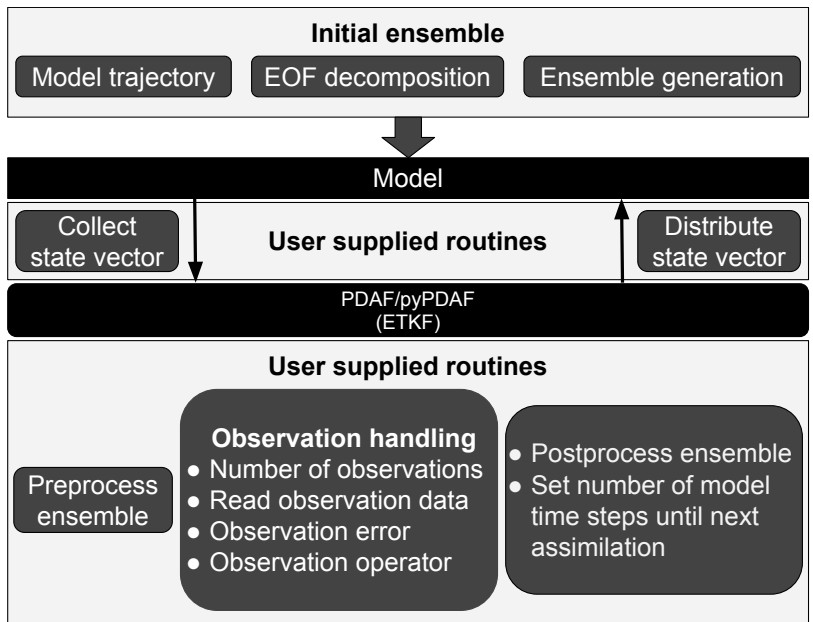

**Figure 1.** A schematic diagram of an online ETKF DA system using (py)PDAF. In the case of an offline DA system, the model can be its restart files.

To ensure that PDAF can be flexibly adapted to any models and observations, it requires users to provide case-specific information. This includes the information on the state vector, observations and localisation. The framework obtains this information via *user-supplied functions* which are external callback subroutines in Fortran. Figure 1 shows a schematic diagram of an online DA system where the ETKF is used. Here, the user-supplied functions connect PDAF with models. Called within the

PDAF routines, these user-supplied functions collect state vectors from model forecasts and distribute the analysis back to the model for the following forecast phase. During the analysis step, user-supplied functions also pre- and post-process the ensemble, handle observations and provide the number of model time steps for the next forecast phase to PDAF. As the user-supplied functions depend on the chosen DA algorithm, other algorithms may require additional functions. For example, the local ensemble Kalman filter (LETKF) requires routines used to handle the domain localisation and 3DVar requires routines for the

adjoint observation operator and control vector transformation. To ameliorate the difficulty in the observation handling, PDAF provides a scheme called observation module infrastructure (OMI). The OMI routines currently support spatial interpolations, direct observation operator, and a diagonal observation error covariance matrix.

In an online DA system, the collection and distribution of state vector is an in-memory data exchange handled by PDAF. It is possible to implement an offline DA system with PDAF where the model in Fig. 1 can be simply replaced by model restart

files while the collection and distribution routines manage the I/O operations.





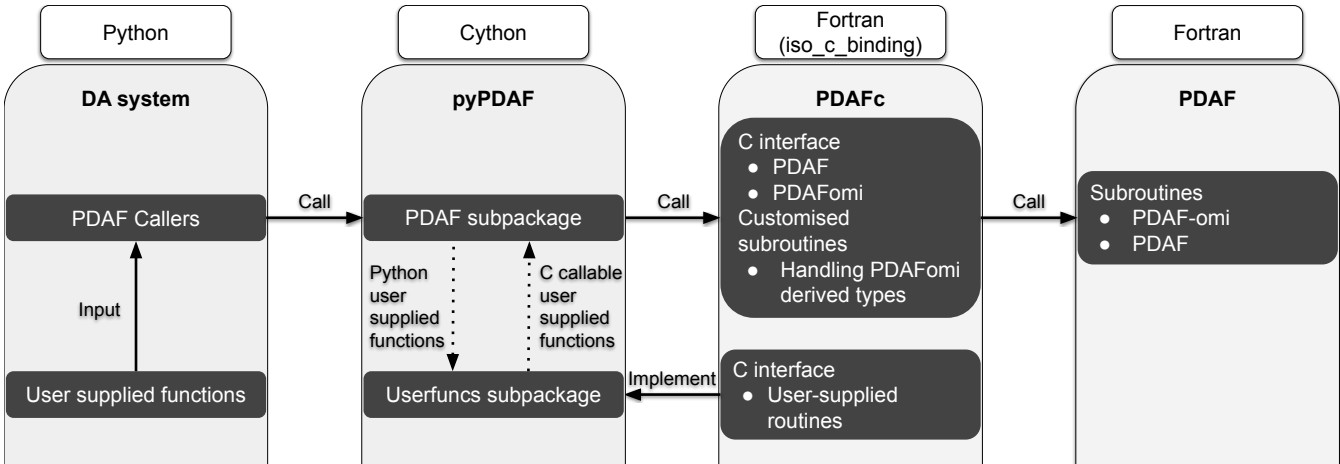

**Figure 2.** An illustration of the design of the pyPDAF interface to the Fortran-based framework PDAF.

## 3.2 pyPDAF

Implementation of user-supplied functions can be laborious in Fortran and typical code development in Python can be less time consuming. Due to the integrated package management, code development in Python can rely on well optimised packages without the need for compilation. For these reasons, a variety of numerical models are implemented in Python (e.g., De Cruz

et al., 2016; Abernathey et al., 2022; McGibbon et al., 2021; Bi et al., 2023). Hence, a Python interface to PDAF allows for designing an online DA system with Python-based models, and also allows for an efficient code development and modifications for a DA system such that users can focus on scientific problems. These features can also be attractive for a prototypical DA system before performing an optimised implementation for high-dimensional Fortran-based models. Generally, it would also be possible to interface a Fortran-based model to pyPDAF, which then interfaces to PDAF. However, this latter approach might

be inefficient due to in-memory copies of large arrays between Fortran and Python.

The pyPDAF package can also be applied for offline DA system, i.e. coupling the model and data assimilation program through restart files, where pyPDAF can be used without the restriction of the programming language of the numerical model. When user-supplied functions are well optimised, this could also be used for complex models as pyPDAF fully supports the parallel features of PDAF using the Message Passing Interface (MPI, Gropp et al., 1994).

As the reference implementation of Python is based on the C programming language (cf. The Python Language Reference), the design of pyPDAF is based on the interoperability between the programming languages C and Fortran using the iso_c_binding module of Fortran. As shown in Fig. 2, the C interface of PDAF is implemented in *PDAFc* which includes essential PDAF interfaces and interfaces for user-supplied functions. Hence, PDAFc could be used independently from pyPDAF as a C interface to the PDAF package. The core of the pyPDAF implementation uses the C-extension for Python (Cython).

Here Python datatypes are converted into C pointers to allow for information exchange between PDAF and pyPDAF. pyPDAF



implements C callable functions which can call user-supplied functions in Python such that PDAF can utilise the user-supplied Python functions.

With the design of pyPDAF, the package is a mixed program of C, Fortran and Python. Moreover, as DA algorithms require high-dimensional matrix multiplications, PDAF relies on the numerical libraries LAPACK (linear algebra package) and BLAS
(basic linear algebra subprograms). These libraries lead to a complex compilation process especially when users could use different operating systems. To fully utilise the cross-platform Python environment, pyPDAF is distributed via the package manager *conda* to provide an out-of-box user experience with pyPDAF where users can use pyPDAF without the need for compiling the package from the source code. Detailed installation instructions can be found at: https://yumengch.github.io/pyPDAF/install.html
pyPDAF allows for the use of efficient DA algorithms in PDAF. However, a DA system purely based on pyPDAF could still be less efficient than a DA system purely based on PDAF. The loss of efficiency is partly due to the user-supplied Python functions and the conversion of data between Fortran and Python objects leading a computational overhead. We will evaluate the loss of efficiency in Sect. 6.3.

## 4   Coupled data assimilation

To demonstrate the use of pyPDAF and PDAF, a coupled data assimilation (CDA) setup is used. In coupled models, information between model components are exchanged during the model forecast at specified time intervals. In WCDA systems, in contrast to the coupled model forecast, each model component performs its own DA without considering the state of other model components. In SCDA systems, the DA system updates the model components jointly where the observations from each model component can affect other model components.

To facilitate the discussion of the effects of SCDA in the numerical experiments performed in Sec. 6, we illustrate the SCDA by a system with two components where each component has only one scalar variable. Here, in order to simplify the equations, observations are assimilated serially which is possible if the observation errors are uncorrelated. We write the two-component state vector is $\mathbf{x} = \begin{pmatrix} x_1 & x_2 \end{pmatrix}^{\mathrm{T}}$. We assume that each component of the state vector is directly observed (i.e. the observation operator is an identity matrix). Thus we have an observation $y_i$ with an error variance of $r_i$ for the $i$-th component. Even though
the ensemble formulation is not used in the discussion here, we assume for the $i$-th model component an inflation factor $\sqrt{\beta_i}$ for the forecast error, or ensemble anomaly in the context of ensemble DA. Thus the covariance matrix is written as

$$\mathbf{P}^f = \begin{pmatrix} \beta_1 p_1^f & \sqrt{\beta_1\beta_2}p_{12}^f \\ \sqrt{\beta_1\beta_2}p_{21}^f & \beta_2 p_2^f \end{pmatrix} \qquad (1)$$

Applying this to the analysis equations of the Kalman filter (e.g., Asch et al., 2016) one obtains the increment of $x_1$ as

$$\delta x_1 = \frac{\beta_1 p_1^f d_1}{\beta_1 p_1^f + r_1} + \frac{p_{12}^{a_1}}{p_2^{a_1} + r_2}\left(d_2 - \delta x_2^{a_1}\right), \qquad (2)$$

where the superscript $f$ represents the forecast, while $a_1$ represents the analysis after assimilating observation $y_1$, $p_i$ is the error variance of the $i$-th model component and $d_i = y_i - x_i^f$ is the innovation of the $i$-th model component, the analysis of $x_2$ after



assimilating $y_1$ is

$$\delta x_2^{a_1} = \frac{\sqrt{\beta_1 \beta_2} p_{21}^f d_1}{\beta_1 p_1^f + r_1},\tag{3}$$

and $p_{12}^{a_1}$ is the cross error covariance between two model components after assimilating $y_1$:

$$p_{21}^{a_1} = p_{12}^{a_1} = \sqrt{\beta_1 \beta_2} p_{12}^f \left(1 - (\beta_1 p_1^f + r_1)^{-1} \beta_1 p_1^f\right) \quad \text{and}\tag{4}$$

$$p_2^{a_1} = \beta_2 \left(p_2^f - (\beta_1 p_1^f + r_1)^{-1} \beta_1 p_{21}^f p_{12}^f\right).\tag{5}$$

The corresponding analysis variance is:

$$p_1 = \beta_1 p_1^f \left(1 - \beta_1 p_1^f (\beta_1 p_1^f + r_1)^{-1}\right) - (p_2^{a_1} + r_2)^{-1} p_{12}^{a_1} p_{21}^{a_1}.\tag{6}$$

The first term in Eq. (2) is the increment due to the WCDA in model component 1, while the second term is the SCDA effect. Thus, if the second term in Eq. (2) becomes zero, the increment is equivalent to a WCDA update for $\delta x_1$ as the observation $y_2$ has no impact on the analysis of $x_1$. Similarly, in Eq. (6), the second term associated with cross error covariance reduces the amount of uncertainty from the WCDA which only contains the first term of the equation. Equation (2) and (6) demonstrate the importance of cross-covariance in the SCDA system. This discussion likewise applies to the analysis update of $x_2$ as a similar equation can be obtained when $y_2$ is assimilated before $y_1$ to update $x_2$.

In the case that not all model components have observations, SCDA can provide an estimate of the un-observed model components based on available observations. For example, when only the first model component is observed, the analysis increment and variance of the un-observed second model component is $\delta x_2^{a_1}$ and $p_2^{a_1}$ as detailed in Eq. (3) and (4). Equation (3) shows that, the DA increment of unobserved variable depends entirely on the cross error covariance. A similar result has been shown under an incremental 4DVar setup by Smith et al. (2020) where they investigated the error covariance matrix using a single observation in each model component. Note that Eq. (3) can also be used to understand multivariate DA and parameter estimations that will not be discussed here.

## 5 Model and DA setup

To demonstrate the application of pyPDAF and to evaluate its performance in a coupled DA setup, the Modular Arbitrary-Order Ocean-Atmosphere Model ([MAOOAM, De Cruz et al., 2016) version 1.4 is coupled with PDAF and pyPDAF. The original MAOOAM model is implemented in Fortran that is coupled directly with PDAF, and a wrapper for Python is developed in this study such that MAOOAM can be coupled with pyPDAF. This means that two online DA systems using Fortran and Python respectively are developed to allow for a comparison between the PDAF and pyPDAF implementation. In these DA systems, a suite of twin experiments are carried out using the ensemble transform Kalman filter (ETKF, Bishop et al., 2001).

### 5.1 Coupled model MAOOAM

The MAOOAM solves a reduced-order non-dimensionalised quasi-geostrophic (QG) equation (De Cruz et al., 2016). Using the beta-plane approximation, the model has a two-layer QG atmosphere component and one-layer QG shallow-water ocean



component with both thermal and mechanical coupling. For the atmosphere, the model domain is zonally periodic and has a no-flux boundary condition meridionally. For the ocean, no-flux boundary conditions are applied in both directions. This setup represents a channel in the atmosphere and a basin in the ocean. The model variables for the two-layer atmosphere are averaged

into one layer. Accordingly, MAOOAM consists of four model variables: the atmospheric streamfunction, $\psi_a$, the atmospheric temperature, $T_a$, the ocean streamfunction, $\psi_o$, and the ocean temperature, $T_o$. The model variables are solved in a spectral space. The spectral basis functions are orthonormal eigenfunctions of the Lapace operator subject to the boundary condition, and the number of spectral modes is characterised by harmonic wave numbers $P$, $H$, $M$ (Cehelsky and Tung, 1987).

We integrate MAOOAM with (py)PDAF with an application of the ensemble transform Kalman filter (ETKF). As shown in
Fig. 1, the key ingredient of coupling MAOOAM with (py)PDAF is the collection and distribution of state vector. In common setups of ocean and atmospheric DA, the observations are available in the physical space. Hence, in the user-supplied function that collects the state vector for pyPDAF (cf. Fig. 1), spectral modes of the model are transformed from the spectral space to physical space using the transformation equation,

$$f(x,y,t) = \sum_{i=1}^{K} c_i(t) F_i(x,y), \tag{7}$$

where $f(x,y,t)$ is any model variable in the physical space, $K$ is the number of modes, $c_i(t)$ is the spectral coefficient of the model variable, $F_i(x,y)$ is the spectral basis function of mode $i$ outlined in De Cruz et al. (2016). In the user-supplied function that distribute the state vector for pyPDAF (cf. Fig. 1), the analysis has to be transformed back to the spectral space to initialise the following model forecast. The inverse transformation from the physical space to the spectral space can be obtained by

$$c_i(t) = \frac{n}{2\pi^2} \int\limits_{0}^{\pi} \int\limits_{0}^{\frac{2\pi}{n}} f(x,y,t) F_i(x,y) dx dy. \tag{8}$$

Here, each basis function corresponds to a spectral coefficient of the model variable. The basis functions are evaluated on an equidistant model grid. The spectral coefficients are obtained via the Romberg numerical integration. To ensure the accuracy of the numerical integration, the number of grid points is $2^k + 1$ with $k \in \mathcal{Z}^+$.

Our model configuration adopts the strongly coupled ocean and atmosphere configuration (36st) of Tondeur et al. (2020) using a time step of $0.1$ time units corresponding to around 16 minutes. Using the notation of $H^{max}x - P^{max}y$ of De Cruz et al.
(2016), the configuration chooses $2x - 4y$ modes for the ocean component and $2x - 2y$ modes for the atmosphere component. This leads to a total of 36 spectral coefficients with 10 modes for $\psi_a$ and $T_a$ respectively and 8 modes for $\psi_o$ and $T_o$ respectively. The model runs on a rectangular domain with a reference coordinate of $(x \times y) \in [0, \frac{2\pi}{n}] \times [0, \pi]$, where $n = 1.5$ is the aspect ratio between the $x$ and $y$ dimensions.

In contrast to Tondeur et al. (2020) who assimilate in the spectral space, we assimilate in the physical space in which the
observations are usually available. To determine the number of grid points for the physical space, a sensitivity experiment was performed to study the transformation error. The experiment shows that when the number of grid points reaches $129 \times 129$, the transformation error becomes negligible and the physical grid points resolve the features in the spectral space. In practice, due to the chaotic nature of the model and long simulation time, the error from the transformation can accumulate which



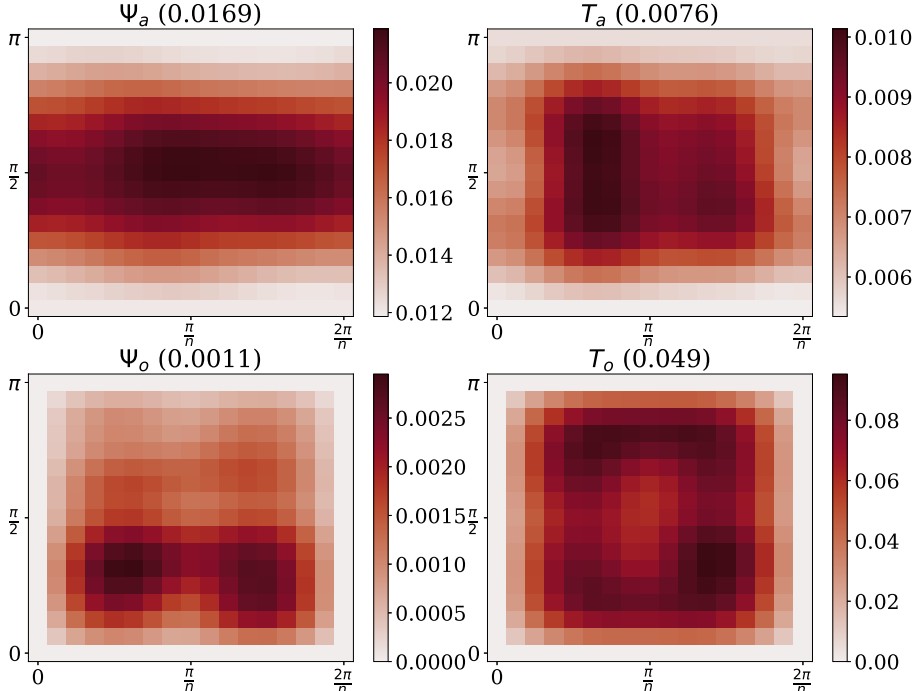

**Figure 3.** The observation error standard deviation fields used for generating the synthetic observations. The spatial mean of the error standard deviation is shown in the bracket.

subsequently leads to model errors. For the sake of efficiency, $129 \times 129$ grid points are chosen. This gives us an order of $10^4$
grid points for each model variable in the state vector for DA.

## 5.2 Experiment design

In a twin experiment, a long model run is considered truth. The model state is simulated with an initial condition sampled in the spectral space which follows a Gaussian distribution, $\mathcal{N}(0, 0.01)$. The DA experiments are started after $10^5$ time steps corresponding to around 277 years of model integration to ensure the dynamical consistency of the model state.

The observations are generated from the truth of the model state based on pre-defined error statistics of the observations. Both atmosphere and ocean observations are sampled every 8 model grid points leading to $17 \times 17$ observations for each model field. The observation error standard deviation are set to $50\%$ and $70\%$ of the temporal standard deviation of the true model trajectory for the atmosphere and ocean respectively. The resulting standard deviation of the atmosphere observations is on a similar magnitude with the ensemble spread of the free run (cf. Fig. 4) while the magnitude of the observation error
in the ocean is typically larger than in the atmosphere in real observing networks. The obtained standard deviation fields are shown in Fig 3. With our chosen model configuration, the highest observation error is in the ocean temperature while the ocean streamfunction shows the least uncertainty due to its slow variability. Meanwhile, the atmospheric processes in MAOOAM





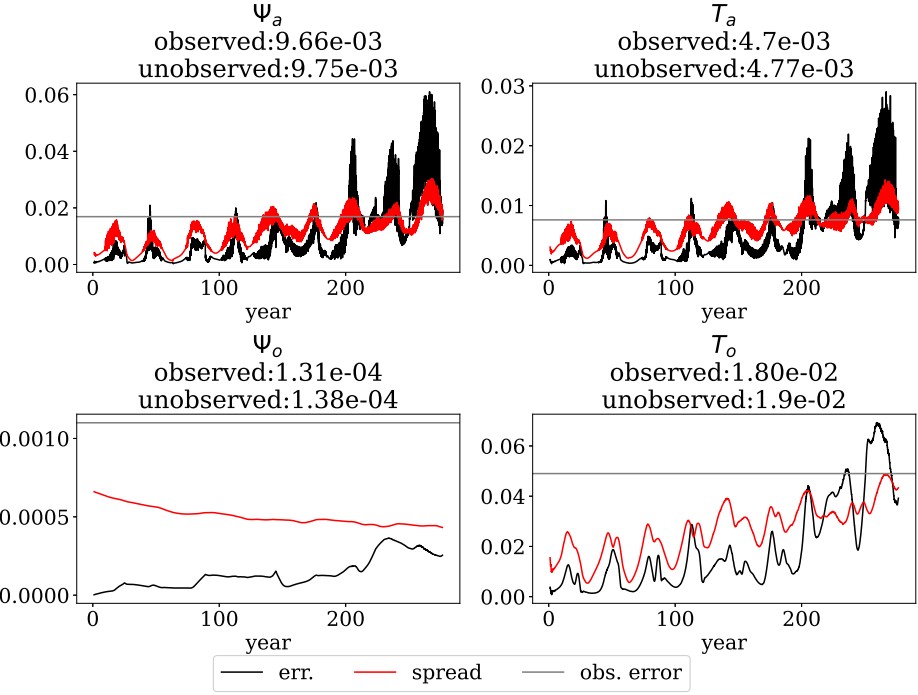

**Figure 4.** The time series of spatial mean of ensemble spread (red), RMSE of the *ensemble mean* with regard to the truth (black) and observation error standard deviation (grey) in the free run neglecting the spin-up period. The atmosphere shows fast variability and oscillatory RMSE of the ensemble mean while the RMSE of the ocean ensemble mean is smooth. The temporal mean of the RMSEs calculated over observed and un-observed gridpoints is also given.

show fast variability and shorter timescale than the ocean. Hence, the ocean observations are assimilated around every 7 days (630 time steps) while the atmosphere observations are assimilated around every day (90 time steps).

As shown by Tondeur et al. (2020), DA in the model configuration using 36 spectral coefficients can achieve sufficient accuracy with an ensemble size of 15 members. In this study, 16 members are used and each ensemble member runs serially with a single process. An ETKF without spatial localisation is used and, without tuning, a forgetting factor of 0.8 is applied to maintain the ensemble spread and ensure a stable DA process.

Using the second-order exact sampling provided by PDAF (cf. Sect. 3.1), the ensemble is generated from a model trajectory
by sampling the modelled truth every 10 days over 100 years after around 1000 years from the beginning of the simulation. This leads to 36 non-zero singular values equaling to the number of spectral modes in the model. The perturbation from the second-order exact sampling could violate the dynamical consistency of the model, so that the ensemble would need to be spun up to reach dynamical consistency. To reduce the spin up time, the initial perturbation is scaled down by a factor of 0.2, 0.15, 0.4 for $\Psi_a$, $T_a$ and $T_o$ respectively. Because the ocean streamfunction has very low variability, its perturbation is unchanged.





The DA experiments are started after 15 days from the beginning of the ensemble generation. In this setup, the forecast error is solely a result of inaccuracy of initial conditions. As shown in Fig. 4, the ensemble spread generally captures the trend and is in a similar magnitude of the model forecast error. This suggests that the forecast uncertainty from the free run ensemble initialised by the second-order exact sampling is able to reflect the forecast errors.

In the free run, the ocean temperature shows the highest uncertainty compared to other model variables. The ocean stream-
function shows a very slow error growth rate. This is also shown by the error and ensemble uncertainty which are two magnitude smaller than those of other model variables. Sensitivity tests (not shown) suggest that an increased error of the ocean streamfunction has a significant impact on the model dynamics consistent with the theoretical discussion given in Tondeur et al. (2020). The error of the atmosphere components shows a wave-like behaviour in time. Tondeur et al. (2020) describe the periods associated with fast dynamics with high and oscillatory errors as active regimes and the periods associated with slow
dynamics with low and stable errors as passive regimes.

## 6   Results

In this section, we evaluate the accuracy of WCDA and SCDA in the MAOOAM-(py)PDAF online DA system using ETKF. To evaluate the computational efficiency of pyPDAF and PDAF, we also compare the wallclock time required by the WCDA and SCDA system. The online DA systems using PDAF and pyPDAF produce quantitatively the same results in all WCDA
and SCDA experiments up to machine precision.

### 6.1   Weakly coupled data assimilation

In WCDA, the error cross-covariances between atmosphere and ocean do not influence the analysis. Instead, the coupling only occurs during the model forecast. This means that the sparse observations only influence their own model component in the analysis step. In this setup, each model component has its own DA system with only two model variables, the streamfunction
and temperature, on the same model grid. The implementation of such an online DA system requires two separate state vectors in each analysis step which is not straightforward with PDAF due to its assumption that each analysis step has only one state vector. (In the case of AWI-CM in Tang et al. (2021), two separate state vectors were obtained by using a parallelization, but here the two model components of MAOOAM are run using the same processor.) This restriction is circumvented by resetting the time step counter in PDAF in our implementation such that even if the assimilation of two state vectors are done by using
PDAF twice, PDAF only counts it as one analysis time step. An alternative approach could be to use the localized LETKF method and define the local state vector as either the atmosphere or ocean variables. The WCDA results are suitable to be a baseline to demonstrate the advantage of SCDA.

Figure 5 shows that the time averaged RMSE of WCDA is smaller than that of the unconstrained free run thus that error growth is successfully controlled. This also demonstrates that the ETKF leads to a converged analysis even though our
observations are less accurate than the forecast at the start of the DA period.



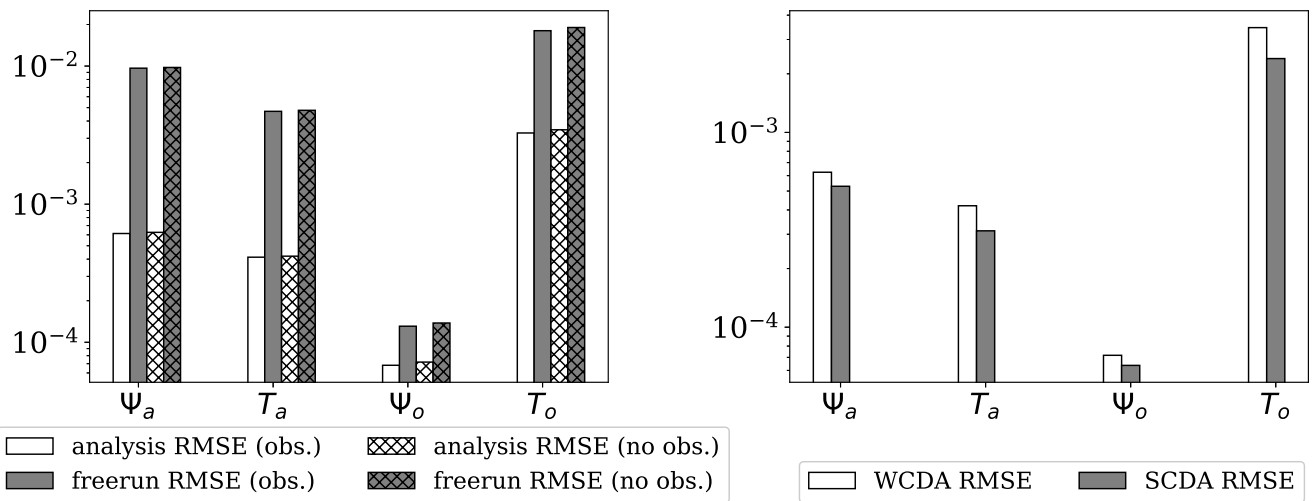

**Figure 5.** Left: The time-averaged RMSE of the analysis using WCDA and free run where the RMSE of the observed, denoted by "obs." in the legend, and unobserved gridpoints, denoted by "no obs.", are compared separately. Right: comparison of RMSEs for weakly and strongly coupled DA for all grid points. The y-axis is plotted in the log-scale and the hatched bars represent the RMSE in the regions without observations.

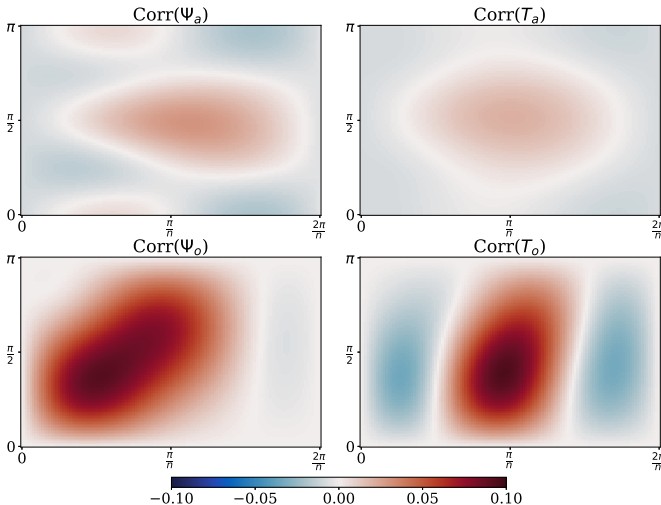

**Figure 6.** The spatial error correlation of each model grid point with the grid point at $\left(\frac{\pi}{n}, \frac{\pi}{2}\right)$ using the ensemble anomaly after 200 years of WCDA.



Similar to the free run, the WCDA results show comparable RMSEs on observed and unobserved grid points even though only a selection of gridpoints are observed. The effectiveness of the DA on the unobserved grid points relies on the spatial error correlations. In the ETKF, the error covariance matrix is estimated by the ensemble anomaly matrix which could be subjected to the sampling error considering our DA system only contains 16 ensemble members. These sampling errors are commonly

controlled by spatial localisation in the ETKF when it is applied to high-dimensional models. Our ensemble DA system shows improved state estimates of the unobserved model grid points using the ensemble-sampled error covariance matrix without any localisation. This is likely caused by the spectral model setup where the model is mostly composed of long waves leading smooth spatial variations and the homogeneous spatial observation network. Figure 6 shows the error correlation of the grid point in the centre of the domain with other grid points computed from the model forecast ensemble at the end of the experiment

period. The spatial correlation field is smooth with long length-scale demonstrating that the ensemble size is sufficient for the system. The smooth error correlation fields shows a wave-like structure with strong positive correlations for regions that are close to the centre of the domain and negative correlations near the domain boundaries. Thus, despite a discretisation using $129 \times 129$ grid points, the effective state dimension is much lower.

### 6.2   Strongly coupled data assimilation

Compared to the WCDA, atmosphere observations influence the ocean part of the state vector and vice versa in the SCDA. This means that the coupling occurs for both the analysis step and model forecast. In this case, the DA system only has one state vector that contains the streamfunction and temperature of both model components. The implementation of an online SCDA system aligns with the design of PDAF and does not require special treatment.

As expected, the SCDA yields lower analysis errors than the WCDA as shown in the right panel of Fig. 5. The improved

analysis in the SCDA in each model component is a result of assimilating observations from the other model component. The effective use of these additional observations relies on the error cross-covariance matrix between model components estimated by the forecast ensemble. The improvements suggest a reliable error cross-covariance matrix in the coupled DA system.

To further understand the effect of the cross-covariances, and the advantage of assimilating observations from the other model component, we further carry out experiments in which only one model component is observed. In the SCDA, the analysis

increment of a model component without observations relies on the error cross-covariance matrix with the model components that have observations. This set up corresponds to Eq. (3) which also shows that the analysis increment is proportional to the inflation $\sqrt{\beta_2}$ applied to the unobserved component. Here, to avoid an excessive analysis increment, $\sqrt{\beta_2}$, is set to one. This is achieved in the post-processing routines as PDAF applies inflation uniformly to the entire state vector by default.

Figure 7 shows the time-averaged RMSE of fields that are smoothed in time by a moving average as a function of the

averaging time-window. The RMSEs of the instantaneous model fields are represented by zero moving average window length. Assimilating observations from the other model component with SCDA can improve the analysis of the unobserved model component. This suggests again that the error cross-correlation between atmosphere and ocean is sufficiently reliable. The atmosphere observations are more effective in controlling the ocean errors than the ocean observations themselves. This shows the necessity to control the errors in the fast changing atmosphere as was discussed by Tondeur et al. (2020). Another possible





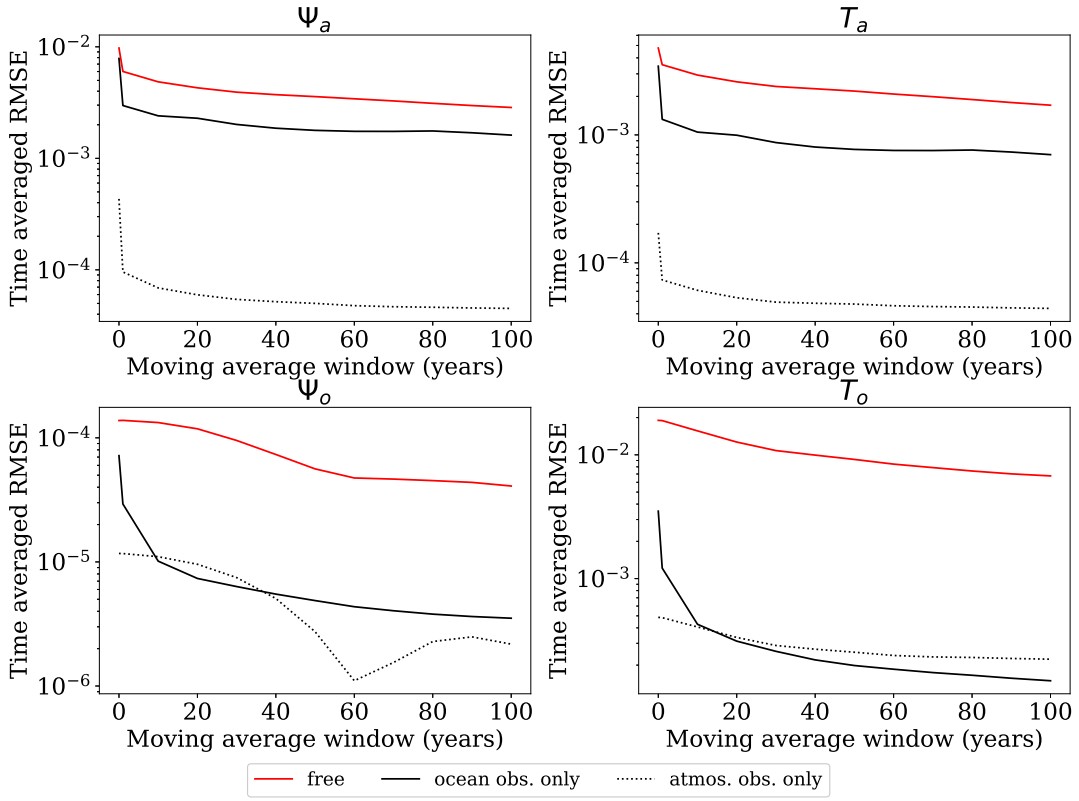

**Figure 7.** Time averaged RMSE when only one model component has observations. The y-axis is in log-scale.

explanation for the effective SCDA of atmospheric observations might be that the ocean observations are less frequent and accurate than the daily atmosphere observations.

As shown in Fig. 7, the assimilation not only improves the instantaneous model fields but also the long-term trend of the atmosphere and ocean climate even though the error dynamics of atmosphere and ocean shows strong time-scale differences in Fig. 4. This means that the ocean dynamics benefit from atmosphere observations even if the transient atmosphere processes

are smoothed by the moving average. Notably, the RMSE of the ocean streamfunction when only atmospheric observations are assimilated does not decrease monotonically with the moving average window length. This could be explained by the fact that the time averaged ocean streamfunction shows periodic features in time and an moving average of $\sim 60$ years leads to a time series of nearly constant streamfunction. This improves the skill of the DA. However, this feature is not captured by the analysis that assimilates ocean observations perhaps due to the large observation uncertainties.

**6.3 Computational performance of PDAF and pyPDAF**

One motivation of developing a Python interface to PDAF is that the efficient DA algorithms in PDAF can be used by pyPDAF while the user-supplied functions can be developed with the ease of Python. However, the user-supplied functions provided by





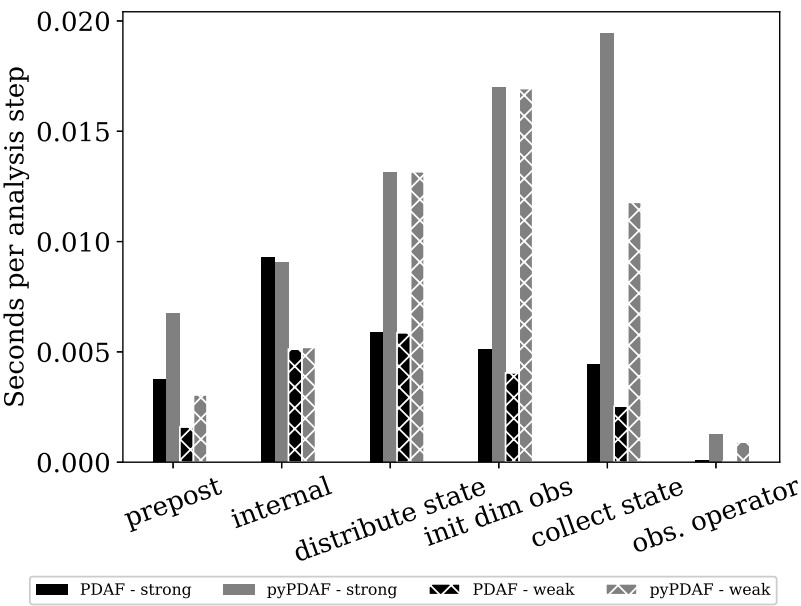

**Figure 8.** Wall clock time of PDAF and pyPDAF for each analysis step averaged over $10^5$ analysis steps over 50 repeated experiments. Only subroutines that use more than $10^{-5}$ seconds for each analysis step are presented. The hatched bars represent WCDA results.

Python are expected to be slower than a pure Fortran implementation. Here we present a comparison of the wall clock time of both PDAF and pyPDAF experiments with standard SCDA and WCDA browken down to the level of subroutines. Each experiment runs $10^5$ analysis steps and each experiment is repeated 50 times. The computation runs on the UK computing facility JASMIN on a node with two AMD EPYC 7402 24-Core processors which has a 2.8GHz frequency. With 16 ensemble members, each member uses a single process for model forecast and the DA is performed serially on a single process. In these comparisons, each state vector has a dimension of $66,546$ in the SCDA and a dimension of $33,273$ in the WCDA. However, as the WCDA computes the analysis separately for each model component, it conducts the DA twice for each analysis step. The number of observations varies as 578 atmospheric observations are assimilated every 90 time steps and a total of 1156 observations of both atmosphere and ocean are assimilated every 630 time steps.

As shown in Fig. 8, the PDAF-internal procedures, which are the core DA algorithm, take nearly the same amount of time for PDAF and pyPDAF in both WCDA and SCDA. As expected, the user supplied routines require more computational time in the Python implementation of pyPDAF than for the Fortran implementation with PDAF. The distribution and collection routines that exchange information between PDAF and models require significantly more computational time in Python than pure Fortran routines. This could be related to the conversion between physical and spectral space. The most prominent differences come from the 'init dim obs' routine which involves the reading of observations from a file and the construction of the relationship between the state vector and observations. Comparatively, the observation operator takes longer time in Python than Fortran, even if it only calls the Fortran PDAF routine with a loop over different observations. In this case, the Fortran



subroutine calls the Python call-back function, which calls a Fortran subroutine. Hence, the difference between PDAF and pyPDAF implementation is partly a result of computational overhead from data exchange between Python and Fortran. The overhead is expected to be more evident with increasing ensemble size as the observation operator is called $N_e + 1$ times. Overall, the time needed for pyPDAF is approximately 2 to 2.5 times longer than that for the Fortran-implementation when using PDAF directly.

**7   Conclusions**

We introduce a Python package, pyPDAF, an interface to the Parallel Data Assimilation Framework (PDAF) and outline its implementation and design. pyPDAF allows for a Python-based DA system for models coded in Python or interfaced to Python. Furthermore it allows for the implementation of a Python-based offline DA system where the DA is performed separately from the model and data is exchanged between the model and DA code through files. The pyPDAF package, which provides an

interface, allows one to implement user-supplied functions in Python for flexible code development while the DA system is still benefiting from PDAF's efficient DA algorithm implementation in Fortran.

Using a CDA setup, we demonstrate that pyPDAF can be used with the Python model MAOOAM. Both strongly coupled data assimilation (SCDA) and weakly coupled data assimilation (WCDA) are demonstrated. Our results confirm that the SCDA performs better than WCDA, and additional observations from other model components can improve the overall performance

of DA using SCDA. We also investigate the scenario where only one model component is observed. In this case, the error cross-covariance matrix is sufficiently reliable for updating the unobserved model variables leading to improved analyses states for both observed and un-observed model variables. We also show that the DA can improve the long-term trend of the model state in the MAOOAM model.

Using the WCDA and SCDA setup, the computational costs of using pyPDAF and a Fortran-only implementation with PDAF are compared. We show that the computational time stays the same for the core DA algorithm executed in PDAF while pyPDAF

yields an overhead in user-supplied functions. This overhead is a result of both the Python implementation and the requirement of data conversion between Python and Fortran. These overheads may be mitigated by a more efficient implementation of the user-supplied functions and data type definitions.

*Code availability.* The Fortran and Python code and corresponding configuration and plotting scripts including the randomly generated initial

condition for the coupled DA experiments are available at: https://doi.org/10.5281/zenodo.11367123. The MAOOAM V1.4 model used for our experiments is available at https://github.com/Climdyn/MAOOAM/releases/tag/v1.4 with a version available at https://doi.org/10.5281/ zenodo.1308192. The Fortran version of the experiment depends on PDAF V2.1 which is released at https://doi.org/10.5281/zenodo.7861829 and can be also found at https://github.com/PDAF/PDAF/releases/tag/PDAF_V2.1 (Nerger, 2023). The source code of pyPDAF is available at https://github.com/yumengch/pyPDAF/releases/tag/v1.0.0 with the exactly same version at https://doi.org/10.5281/zenodo.10950130.



*Author contributions.*    YC coded and distributed the pyPDAF package, conducted the experiments, performed the data analysis, and wrote the paper. AL and LN both contribute to the conceptual experiment design and the paper writing.

*Competing interests.*    The authors declare that they have no conflict of interest.

*Acknowledgements.*    The authors acknowledge the UK National Environment Research Council's support for the National Centre for Earth Observation (Contract Number: PR140015).



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
