# Peer review of "A Python interface to the Fortran-based Parallel Data Assimilation Framework: pyPDAF v1.0.2"

_EGUsphere, 2024_

## Author Comment (AC1)

**Reviewer 1:**

I quickly reviewed this paper and found that critical issues exist as follows:

**Answer:** We thank for the reviewer for their feedback.

1. Incorrect terminology even in abstract (e.g., model analysis)

   **Answer:** Thank you for bringing up this issue. However, the term "model analysis" can be found in at least a few publications, e.g.:

   • Liu, X., Yao, J., Wu, T., Zhang, S., Xu, F., Zhang, L., et al. (2021). Development of coupled data assimilation with the BCC climate system model: Highlighting the role of sea-ice assimilation for global analysis. Journal of Advances in Modeling Earth Systems, 13, e2020MS002368. https://doi.org/10.1029/2020MS002368

   • Rodell, M., and Coauthors, 2004: The Global Land Data Assimilation System. Bull. Amer. Meteor. Soc., 85, 381–394, https://doi.org/10.1175/BAMS-85-3-381.

   This terminology can also be found in the website of operational centres. For example, the term is used in the caption of Fig. 2.5.1 of in `https://confluence.ecmwf.int/display/FUG/Section+2.5+Model+Data+Assimilation%2C+4D-Var` (last accessed 24 July 2024) of the European Centre for Medium-range Weather Forecasts (ECMWF). To this end, we are confident that the term is correct. We also checked the manuscript and didn't find other potentially incorrect terminology.

2. No explanation of math symbols even in Eq. (1) (e.g., $p_i$)

   **Answer:** We kindly remind the reviewer that we defined $p_i$ after Eq. (2). Based on the suggestion from the second reviewer, we have decided to remove this section. We have checked other mathematical symbols in the manuscript and are confident that everything is defined.

3. Larger forecast/analysis RMSEs than the prescribed observation errors in Fig. 4 (i.e., filter divergence occurs)

   **Answer:** We apologise for the confusion. As described in the caption, Fig. 4 is the free run ensemble without any assimilation being applied. We modified the figure to include both the freerun and SCDA experiment. This is consistent with the conclusion by Tondeur et al. (2020).

Therefore, there is a high possibility that this paper does not satisfy the standards of international journals.

**Answer:** We regret to hear the reviewer's opinion. However, we believe that by providing a new Python interface to PDAF and presenting a robust set of experiments to demonstrate its performance, this manuscript is a useful contribution to the international community.

**Reviewer 2:**

**Summary:**

The authors introduced a new Python interface for the PDAF software. The idea of implementing a Python wrapper for a sophisticated Fortran library is much welcomed, since coding in Python is much easier and it will allow more rapid development especially for new models written in Python. However, I found the authors missed the opportunity to convince the readers in this paper that pyPDAF will provide them with tools to quickly implement DA in their Python models. Instead, a lot of focus is put on the details such as second order exact sampling (in SEIK) and comparing weakly and strongly coupled DA. As a user, I only see some hints of what I need to do to use pyPDAF to build a DA system, but I don't know exactly how after reading the paper. With the benchmarking results the authors seem to prefer the Fortran-based PDAF anyway. The test case dimension is very small comparing to typical application, so there is a question whether the Python-Fortran type conversion overhead is significant if pyPDAF is applied to large-dimensional real models. Overall, I feel that the authors didn't promote the new software with convincing enough results.

**Answer:** We thank the reviewer's constructive advice and we revised the manuscript carefully. In our revised manuscript, we show that pyPDAF is usable for high-dimensional systems with limited overhead compared to the Fortran implementation. We reduce the description of second order exact sampling. We added one section that describes the design and implementation of a DA system using pyPDAF. We also revised the subsection that compares the computational efficiency of pyPDAF and PDAF using different dimensions of the state vector up to a large grid size of $2049 \times 2049$ grid points.

**Major issues:**

1. While Figs 1 and 2 provide the overview of software architectures, the readers will not understand exactly what's needed from their side to build a typical DA system fully based on Python. Can you provide an example with a typical cycling DA experiment setup (initial ensemble generation, ensemble forecasts, compute observation priors, collect/distribute state, assimilate algorithm, etc), and highlight the functions where users can either code their own version, or use something readily available from PDAF

core? The OMI is also quite confusing, what functions does it provide? I feel that you have omitted these details because they are available either in PDAF documentation or in previous papers, but listing the details here can convince readers more effectively that it is "easy" to build a system using pyPDAF.

**Answer:** To address the issue of lack of example, we add a new section (Section 3.3): "Construction of data assimilation systems using pyPDAF" where we introduce the necessary user-supplied functions and their intentions for the local ensemble transform Kalman filter as an example. In this section, we also introduce some OMI functionalities and describe the need for additional user-supplied functions without OMI.

2. There are excessive details on the second-order exact sampling which I found not relevant for the topic of this paper. You can use SEIK to demonstrate that the software works, but there are many other options in PDAF that also available. You should definitely save the page limit for something more important. The same thing applies to the comparison between the weakly and strongly coupled DA. The main theme of the paper should be 1. providing the details of how pyPDAF is designed, 2. what's its advantage over other Python DA prototypes, 3. how to use it to build a Python DA system, then 4. some results from a test case. Currently number 2 and 3 are quite missing.

**Answer:** The second order exact sampling is used to generate the initial ensemble for experiments in this manuscript. As it is not the central of this manuscript, we reduce the content of the second order exact sampling in the revised manuscript. The weakly and strongly coupled DA is taken as an example to demonstrate the use of pyPDAF. However, we recognise that the original Sect 6.1 and 6.2 were too verbose and merged these sections into one and removed some content. We also removed the original Section 4 that introduce coupled DA. To address the point 2, in introduction, we added a discussion on the pyPDAF and other Python-based DA packages in terms of its applicability, efficiency and limitations:

"For the application of DA in Python, DAPPER provides a variety of DA algorithms for twin experiments using low-dimensional Python models. The Ensemble and Assimilation Tool, EAT (Bruggeman et al., 2024) was proposed to set up a 1D ocean-biogeochemical DA system, which is a wrapper to a Fortran data assimilation system based on PDAF including the 1D ocean-biogeochemical model, GOTM-FABM. There are also Python packages designed mainly for pedagogical purposes in low-dimensional systems such as openDA (Ahmed et al., 2020) and filterpy (filterpy PyPI, last access: 2024-08-29). For high-dimensional applications, there are efficient implementations of DA packages such as HIPPYlib by Villa et al. (2021) and ADAO (SALOME The Open Source Integration Platform for Numerical Simulation, last access: 2024-08-29), but HIPPYlib does not have a focus on ensemble data assimilation approaches whereas ADAO provides various ensemble DA methodologies but it has no support for the localisation used in weather and climate applications. More recently, NEDAS (Ying, 2024) was recently introduced for offline ensemble DA in high-dimensional climate applications but it currently only supports limited DA algorithms."

Point 3 is addressed in the new Section 3.3 of "Construction of data assimilation systems using pyPDAF".

3. Efficiency benchmark in Figure 8 is performed for a really small model ($129 \times 129 \times 3$) and ensemble size (16), the number of observations also really small ($17 \times 17$?) What the figure is conveying is that coding in Python introduced additional overhead and the resulting software is much slower, therefore using the Fortran code is better? I would argue otherwise: for the test case chosen here, if it only take  0.02 seconds per cycle, I would much prefer coding in Python since it will save me days of work compared to coding in Fortran. It doesn't matter if its 0.02 seconds or 0.005 seconds, they are trivial compare to the "human overhead".

You miss the opportunity here to demonstrate that pyPDAF is a good alternative to PDAF: it should be close to the PDAF efficiency despite the Python-Fortran conversion overhead, but much favorably reducing human overhead in coding. We all know how the ETKF scales with problem sizes, it is roughly $O(Ne^3 + Ne^2 Nobs)$ per state variable. Arguably if the problem size is much larger, and computation spent on the DA problems increase, the overhead of doing the Python-Fortran object conversion can become trivial in propotion to the whole cost. Isn't that a better story to tell?

**Answer:** The original intention of the efficiency benchmark is purely to investigate the potential efficiency loss in Python and Fortran implementation. We added tests with increasing spatial resolutions up to $2049 \times 2049$ number of grid points for global filter. The results show encouraging results for using pyPDAF with a global filter. The pyPDAF performance is similar to PDAF even when the size of the state vector is at a magnitude of 16 million where the overhead becomes less significant compared to the intensive numerical computation.

We also added one test with domain localisation. In this case, the computational cost increases significantly. The domain localisation also introduces increased overhead for some user-supplied functions as they're executed repetitively for each local domain. To address this issue, we introduced a new "PDAFlocal" module in PDAF such that the overhead in the domain localisation is reduced and pyPDAF can achieve similar efficiency per analysis step compared to the PDAF implementation.

**Minor issues:**

1. Line 51, "The Python tool only has a Python interface to a few PDAF routines", do you refer to EAT? and what's the difference between this "Python interface" and pyPDAF?

**Answer:** We apologise for the clarity issue. The tool EAT is not designed as a PDAF interface. Instead, it is a Python tool which is a wrapper of a Fortran DA system based on PDAF. We rephrased the sentence as '... which is a wrapper to a Fortran data assimilation system based on PDAF including the 1D ocean-biogeochemical model.'

2. Line 54, "...can facilitate code development thanks to ...", maybe you mean "easier code development", because it is still possible to write Fortran code only a bit more time consuming.

   **Answer:** We add 'easier' in the sentence.

3. Line 56, "...written onto a disk", do you mean the model restart files?

   **Answer:** We rephrase the sentence as: "For offline DA systems, DA is performed utilising files written onto a disk, e.g., model restart files."

4. Line 65, "can improve dynamically balanced analysis", need a bit rephrasing, how about "can can improve dynamical balance in the analysis"

   **Answer:** This is rephrased.

5. Line 77, "This allows for an embarrassingly parallel... does not increase with the ensemble size", this statement is a bit too general. The embarrassingly parallel variant is the LETKF where each state variable can be analyzed separately. Some other EnKF variants is more tricky to parallelize. Also, in ETKF the computational cost does increase with ensemble size $O(Ne^3)$, I guess you mean at the end of the statement "does not increase with state dimension"

   **Answer:** We agree that this statement does not apply to all ensemble DA methods, so we rephrased the sentence to "The ensemble model forecast allows for an embarrassingly parallel implementation which means that, with sufficient computational resources, the wall clock computational time of the forecast does not increase with the ensemble size.". However, it is worth noting that the Monte Carlo methods are theoretically embarrassingly parallel. This is the case for stochastic ensemble Kalman filter or ensemble variational methods. We recognise that the PDAF parallelisation takes advantages of the domain localisation and this is mentioned in the section "In particular, the domain localisation is tailored for efficient parallel implementations that are commonly used in high-dimensional DA systems.".

6. Line 81, this sentence is repeat of the first sentence.

   **Answer:** We removed "under the Gaussian assumption" in the second sentence.

7. Line 86, "In practice computational resources limit the feasible ensemble size", are you referring to the fact that model forecasts cost a lot so that one cannot run huge ensemble forecasts?

   **Answer:** Yes. We added "... due to the cost of model forecasts" here.

8. Line 105, 108, you've defined PDAF earlier, so why repeating the full name over and over?

   **Answer:** We removed the full name.

9. Line 110, smoother, 3DVar and other non-linear filters, these are not introduced earlier, either mention these in the introduction, or adding some references here.

   **Answer:** We added relevant references.

10. Lines 113-120, why is the detail on second-order exact sampling provided here. PDAF not only provides SEIK but also a lot of other filter types, the SEIK-specific details should be moved to later maybe in the experiment design.

    **Answer:** The second-order exact sampling is the default method provided by PDAF for ensemble generation (see: `https://pdaf.awi.de/trac/wiki/EnsembleGeneration`), and we use this to generate initial ensemble. This sampling method is not limited to the SEIK filter, but was found to be very efficient in previous studies. As discussed in the major comments, we shortened the second-order exact sampling description. We keep the description in place as we consider it to be part of PDAF features instead of an experimental choice.

11. Line 131, the OMI is introduced here in one sentence, can you provide more details. How does the user utilize the functions provided in OMI? Do they import them through the pyPDAF interface or do they have to write their own Cython code to call their Fortran versions?

    **Answer:** The functionalities are added in place and all coding is in Python so that no Cython code has to be written by a user (see response in point 13 as well) .

    "The OMI routines handle the processing of observation vectors and error covariance matrix used by DA algorithms, provide support for the complex distance computation used by localisation. In the current version of PDAF V2.3, it also supports spatial interpolations on structured and unstructured grid, direct observation operator, and a diagonal or non-diagonal observation error covariance matrix. One can implement PDAF without OMI, but additional functions would be required.".

12. Line 134, "replaced by model restart files", this is not trivial to implement, how exactly can this be done? Since this paper only compared two online DA systems using PDAF and pyPDAF, I'm not sure implementing the offline DA is even relevant here.

    **Answer:** We agree that implementing I/O routines for the restart files can be a non-trivial task. We briefly mentioned that "...while the user-supplied collection and distribution routines manage the I/O operations of these restart files." Considering that offline DA

can potentially be an important use-case of pyPDAF, we keep this sentence and add: "Offline DA implementation is a crucially supported feature of PDAF and a potentially important use-case for pyPDAF, but we will not discuss it in detail for the sake of brevity."

13. Figure 2: There are several levels of "user supplied" functions, in both Python and the C interface. This is confusing. As a user using pyPDAF, do I need to code in Python and compile the package, or do I have to also write Cython code? Or, is it just two options?

    **Answer:** We add the following sentences: "pyPDAF is designed so that a DA system can be coded purely in Python including the user-supplied functions and function calls to algorithms implemented in PDAF. The interface to PDAF is provided through functions implemented using Cython, which provides the interface for calls from Python. Thus, the pyPDAF package itself is a mixed program of C, Fortran and Python.", and added sentences in the caption of Fig. 2: "Here, only the Python component is exposed to pyPDAF users, and the Cython and Fortran implementations are internal implementations of pyPDAF"

14. Line 138, "Due to" → "Thanks to"

    **Answer:** Changed.

15. Line 141, "...also allows for an efficient code development and modifications..." this sentence needs some work. I get the first half that pyPDAF allows you to build an online DA system for a Python model. But why does the second half relate to the first half?

    **Answer:** We changed the sentences to: "Hence, a Python interface to PDAF allows the design of an online DA system with such Python-based models. These range from low-dimensional toy dynamical systems to high-dimensional weather and climate systems. Compared to a Fortran-coded DA system, a Python DA system can be implemented efficiently and allows for easier modifications such that users can focus on scientific problems."

16. Line 143, "...before performing an optimised implementation for high-dimensional Fortran-based models", there are Python models that are high-dimensional with well optimised numerics, I don't see the point here why using pyPDAF for a high-dimensional system is not possible now for a prototypical system.

    **Answer:** We agree with the reviewer that pyPDAF can be used with high-dimensional Python models. We changed the sentence mentioned in comment 15 to "Hence, a Python interface to PDAF allows for designing an online DA system with such Python-based models. These ranges from low-dimensional toy dynamical systems to high-dimensional weather and climate systems."

17. Line 149, "pyPDAF fully supports the parallel features", can you provide more details how the MPI featuers are utilized in the Python interface, in Fig. 2 is every function run with MPI, is the Python code run with mpi4py?

    **Answer:** Following the sentence, we added "The application of pyPDAF in high-dimensional models can also be shown by its support of the parallel features of PDAF, which use the Message Passing Interface (MPI, Message Passing Interface Forum, 2023). For this, a pyPDAF DA system relies on the "mpi4py" package for MPI support. The pyPDAF system can also support shared memory parallelisation in PDAF when built with OpenMP. "

18. Figure 4: the error time series seems to be not reaching steady state, it keeps increasing and there is a sign of exponential growth towards the end. Is the filter actually stable in time?

    **Answer:** We apologise for the clarity of the figure. As suggested in its caption, Figure 4 was a time series of the RMSE and ensemble spread of free run that is not constrained by DA. To avoid confusion, we added RMSE and ensemble spread of the SCDA results as a comparison with revised figure caption as well.

19. Line 265, "16 members...ETKF without spatial localization", given the results in Fig. 4 maybe you want to add some localization to stablize the filter.

    **Answer:** As indicated by the results in the new Figure 4, the global filter leads to stable ensemble spread and RMSEs. This is consistent with the results by Tondeur et al. (2020).

20. Line 278, this is not true for the final 100 years, errors are larger than spread.

    **Answer:** We'd like to stress that the free run ensemble spread follows the trend of RMSEs and it is difficult to attribute the low spread to the initial ensemble considering the long model run. We added: "even though the spread is lower than the RMSE after 200 years"

21. Figure 8: you didn't provide information on all the subroutines, what does "init dim obs" mean?

    **Answer:** We change the name of "init dim obs" to "OMI setup", which hopefully is clearer than "init dim obs". We add an explanation of "init dim obs" in Section 3.3:

    "The primary purpose of the function is to obtain the dimension of observation vector, $dim\_obs$ at the current time step as implied by its name. In this function, one has to provide further observation information to OMI."

**References**

Ahmed, S. E., Pawar, S., and San, O.: PyDA: A Hands-On Introduction to Dynamical Data Assimilation with Python, Fluids, 5, https://doi.org/10.3390/fluids5040225, 2020.

Bruggeman, J., Bolding, K., Nerger, L., Teruzzi, A., Spada, S., Skákala, J., and Ciavatta, S.: EAT v1.0.0: a 1D test bed for physical–biogeochemical data assimilation in natural waters, Geoscientific Model Development, 17, 5619–5639, https://doi.org/10.5194/gmd-17-5619-2024, 2024.

filterpy PyPI: `https://pypi.org/project/filterpy/`, last access: 2024-08-29.

Message Passing Interface Forum: MPI: A Message-Passing Interface Standard Version 4.1, URL `https://www.mpi-forum.org/docs/mpi-4.1/mpi41-report.pdf`, 2023.

SALOME The Open Source Integration Platform for Numerical Simulation: `http://www.salome-platform.org/`, last access: 2024-08-29.

Tondeur, M., Carrassi, A., Vannitsem, S., and Bocquet, M.: On temporal scale separation in coupled data assimilation with the ensemble kalman filter, Journal of Statistical Physics, 179, 1161–1185, https://doi.org/10.1007/s10955-020-02525-z, 2020.

Villa, U., Petra, N., and Ghattas, O.: HIPPYlib: An Extensible Software Framework for Large-Scale Inverse Problems Governed by PDEs: Part I: Deterministic Inversion and Linearized Bayesian Inference, ACM Trans. Math. Softw., 47, https://doi.org/10.1145/3428447, 2021.

Ying, Y. M.: nansencenter/NEDAS: v1.0-beta, Zenodo [code], https://doi.org/10.5281/zenodo.10525331, 2024.

---

## Author Response (AR2)

**1 Reviewer 1**

**General comment**

This study evaluates the availability of the ensemble Kalman filter Python package, pyPDAF, for high-dimensional systems by comparing its computational performance with the Fortran-based PDAF. The authors implemented both packages using a simple quasi-geostrophic atmosphere-ocean coupled model. However, while the study aims to demonstrate pyPDAF's applicability to high-dimensional systems, the model's dimensionality  $(10^4 - 10^6)$  is far smaller than that of operational high-dimensional systems, which typically have  $10^8 - 10^9$  grid points (see Comment #2).

Additionally, the ensemble size used in this study is only 16, which is insufficient for a system with dimensions of  $10^4 - 10^6$ . For instance, the 40-variable Lorenz-96 model with ETKF and LETKF requires an ensemble size of at least 40 and 10, respectively, for reliable results. The small ensemble size chosen by the authors undermines the validity of their conclusions regarding pyPDAF's performance. It appears that the authors selected this ensemble size to avoid the computational expense of eigenvalue decomposition for the ensemble-size square matrix in ETKF. However, operational systems commonly use ensemble sizes of  $10^2$ , making it crucial to assess the sensitivity to ensemble size in this study (see Comment #18). The study also suffers from additional shortcomings, including:

- Inaccurate or misleading terminologies (see Comment #1).
- Unrealistic experimental settings, particularly in the choice of observation errors and timescale definitions (see Comments #14-16).
- Inconsistent presentation and substandard English writing quality, which hinder readability and clarity (see Minor Comments).

Given these significant issues, this manuscript does not meet the standards required for publication in an international journal. To address these concerns, the authors need to:

- 1. Align the model dimensionality and ensemble size with the stated objectives.
- 2. Improve the experimental design, particularly in terms of observation errors, timescales, and methodological rigour.
- 3. Correct terminological inaccuracies and enhance the clarity of descriptions.
- 4. Substantially improve the quality of English writing and presentation.

Without these major revisions, the study's claims and findings cannot be considered robust or scientifically valid. The manuscript is not suitable for publication in its current form.

**Answer:** We thank the reviewer for providing the critical review.

We regret the misunderstanding that the reviewer believes the primary goal of the manuscript is to evaluate the availability of EnKF in pyPDAF for high-dimensional systems. We would like to stress that the primary goal of this paper is to introduce pyPDAF to the DA and wider geoscience community regardless of the dimensionality of their applications.

The statement on 'high-dimensional' appears subjective given that highest-dimensional operational systems can hardly serve to define what a high dimension for data assimilation is; see our responses to major comment #2. Additionally, we did not claim that pyPDAF is aimed at operational data assimilation.

However, we agree that the evaluation of the computational performance of the pyPDAF package using ETKF is an important aspect of this manuscript which can be strengthened by increased ensemble size. Therefore, we add experiments with ensemble sizes of 64 and  $128 \text{ on } 257 \times 257 \text{ grid points}$ .

The assumption that we used the rather small ensemble size of 16 to "avoid the computational expense of eigenvalue decomposition for the ensemble-size square matrix in ETKF" is incorrect. Since the eigenvalue decomposition is computed in the compiled Fortran-part of the program provided by PDAF, the execution time will be the same for the native PDAF and the pyPDAF cases. Actually, with a larger ensemble more work is done in the compiled PDAF routines. Thus, increasing the ensemble size would decrease the relative time overhead of the Python-coded part, which is demonstrated in our added experiments with larger ensemble sizes.

The claim that an ensemble size of at least 40 is required for reliable results with the ETKF is inconsistent with existing evidence. In fact, with sufficient inflation, an ensemble size of 16 is sufficient to obtain a stable analysis for 40-variable Lorenz 96 model without any localisation, e.g., see Figure 3 in Bocquet (2011). Even though increased ensemble size can produce better analysis, the improvements are typically marginal. It is also known that one cannot extrapolate from the required ensemble size for the Lorenz 96 model to realistic models. There are, e.g. successful applications of DA in the ocean with very small ensembles (e.g. Yu et al. (2025) using an ensemble size of 8). For the MAOOAM model used in this study, the required ensemble size was examined by Tondeur et al. (2020). Their Figure 7 shows that even an ensemble size of 7 states is sufficient for the configuration we used. The dynamics of our model setup remain unchanged, as grid points are merely a transformation from spectral coefficients.

The responses to the listed shortcomings can be found in comments #1, #14-16 and minor comments. Again, we would like to stress the objective of this manuscript is to introduce a new Python tool for data assimilation implementation and the experiments are designed to demonstrate the software capability instead of scientific aspects of data assimilation methodology and applications.

**Major comment**

- 1) L10 and elsewhere: Please remove the term "model" from phrases like "model forecast" and "model analysis." While such phrasing may appear in other papers, it is likely to be an incorrect use of terminology.
  - Answer: We agree that "model analysis" is less common. However, we keep the "model forecast" as it is quite commonly used.
- 2) L15 and elsewhere: Since operational centers use data assimilation systems with a dimensionality as high as  $10^8 10^9$ , the idealized data assimilation system employed in this study, with a dimensionality of  $10^4 10^6$ , cannot be considered high-dimensional.
  - Answer: We kindly disagree with the argument that a dimension  $O(10^6)$  cannot be high dimensional because operational assimilation systems can have "a dimensionality as high as  $10^8-10^9$ ". In particular, comparing to the systems with the highest dimensions used today does not exclude that a dimension of  $\mathcal{O}(10^6)$  is already high dimensional and there are also operational systems, e.g. in the ocean, which are closer to a dimension  $10^6-10^7$ . Nevertheless, we acknowledge that the term "high-dimensional" is subjective.
- 3) L19–20: In the data assimilation (DA), the description that "observations constraint forecast and results in analyses" would not be accurate. DA combines simulations and observations using dynamical systems theory and statistical methods. This process provides optimal estimates (i.e., analyses), enables parameter estimation, and allows for the evaluation of observation networks. These explanations would be more appropriate.

Answer: Changed.

- 4) L34: The term "user-supplied functions" in Section 1 is unclear, although the authors may provide a detailed explanation later.
  - **Answer:** We have adapted our manuscript as follows: "In this generic framework, DA methods accommodate case-specific information about the DA system through functions provided by users including the model fields, treatment of observations, and localisation. These functions are referred to as *user-supplied functions*."
- 5) First and second paragraphs in Section 2: The Kalman filter can be derived as a minimum variance estimator without assuming Gaussian distributions. However, in most cases, the ensemble Kalman filter (EnKF) assumes Gaussian distributions in the forecast fields, leading to the analysis field following the Gaussian distribution. For non-Gaussian assimilation, such as rainfall, transformations like the logarithm function might be applied. The authors' discussion on ensemble data assimilation appears insufficient. Since detailed discussions on data assimilation methods are not the focus of this study, it is suggested to remove these relevant descriptions.

**Answer:** Following reviewer #3's comments, the section is significantly shortened and reformulated such that the primary purpose of this section is to provide information on available DA algorithms in PDAF.

We agree that the Kalman filter (KF) shares similar properties as a minimum variance estimator. However, the Kalman filter was originally proposed as a Bayesian estimator which specifically assumes Gaussianity. Also, due to the need for the error covariance propagation, the KF is almost always motivated by Gaussian random noises.

Although treating non-Gaussian variables is not the focus of this study, we recognise that it is useful to mention that pyPDAF can perform Gaussian anamorphosis. In Sect. 2.4, we add the following sentences: "For example, as mentioned in Sect. 2.2, optimal state estimation is achieved by ensemble-based Kalman filters under a Gaussian assumption. The state vector collection and distribution function can be used to perform Gaussian anamorphosis where non-Gaussian variables can be transformed to Gaussian variables (Simon and Bertino, 2012)."

**6 - 8) L92-94:**

- Please cite two typical EnKFs: the Ensemble Adjustment Kalman Filter (EAKF; Anderson et al. 2001) and the Ensemble Square Root Filter (EnSRF; Whitaker and Hamill 2002).
- Please remove the mention of the deterministic ensemble Kalman filter (DEnKF), as it neglects quadratic terms such as **KRK**T in its derivation, and therefore is no longer considered an EnKF.
- Please specify which EnKF and data assimilation (DA) methods are included in PDAF and pyPDAF.

**Answer:** The section mentioned in this comment is now used to describe available DA algorithms in PDAF. Though EAKF and EnSRF are important DA algorithms, they are not yet implemented in PDAF yet. We added the following sentence: "Other typical filtering algorithms, not implemented in current releases, such as ensemble adjustment Kalman filter (EAKF, Anderson, 2001) and ensemble square root filters (EnSRF, Whitaker and Hamill, 2002) are planned to be included in future releases."

The reference to DEnKF is removed as it is not implemented in PDAF. However, it is worth noting that the DEnKF was proposed to mitigate the potential degeneracy of ensemble filters (ensemble collapse) while avoiding adding perturbations to observations. The update of the error covariance matrix is described as a first order approximation in a Taylor expansion. The method loses its value if it is not considered EnKF. We are not aware that the scientific community agrees on this view.

The KF variants and other DA algorithms provided by PDAF and pyPDAF are listed in Sect. 2.2.

9) L123–124: Even in twin experiments and Observing System Simulation Experiments (OSSEs), true values are never used to generate forecast ensembles.

**Answer:** Sorry for the mistake. We removed this description. In fact, PDAF can generate an ensemble using any model trajectory.

10) L148–149: Since the amount of work required to implement additional Fortran and Python code depends on the users' skill level, this sentence may not be appropriate.

**Answer:** We add "Depending on the users' programming skills..."

11) L301–303: Please clarify why the transformation errors decrease as the number of grid points increases.

**Answer:** We removed these discussions for the sake of brevity of the experimental setup. The Romberg integration is based on repeatedly applying the trapezoidal rule and using Richardson extrapolation to cancel out the leading error terms. The accuracy of the numerical integration depends on the number of grid points and the spatial resolution with an error of  $\mathcal{O}(n^{-2\log_2 n})$  where n is the number of grid points.

We add: "The accuracy of the numerical integration depends on the spatial resolution and the number of grid points with an error of  $\mathcal{O}(n^{-2\log_2 n})$  where n is the number of grid points. Our experiments suggest that the numerical integration error is negligible once we have  $(2^7 + 1 \times 2^7 + 1) = (129 \times 129)$  grid points."

12) L306–307: The minimum required ensemble size depends on various factors, such as sampling errors and system characteristics. Please provide a detailed explanation of how the unstable subspace dimension influences the ensemble size.

**Answer:** The original sentence was removed for the sake of brevity. In response to the comment, without random model error, the rank of the error covariance matrix of the ensemble Kalman filter converges onto the unstable-neutral subspace of a linear dynamical model as proven mathematically by Bocquet et al. (2017), which is subjected to the observability condition. This might be related to the system characteristics as mentioned by the reviewer.

The optimal ensemble size was studied in Tondeur et al. (2020); Bocquet (2011) for MAOOAM used in this study and Lorenz 96 respectively. In particular, Bocquet (2011) investigated the case with different inflation factors used to deal with sampling errors.

13) L318–321: The descriptions of the amplitude of the observation error variance are inconsistent.

**Answer:** It is unclear what inconsistency is suggested by the reviewer. We shortened the description following the suggestions from reviewer 3.

14) Subsection 4.2: Please clarify why different variables with different units can be directly compared. Non-dimensional temperature and stream function are typically normalized in a quasi-geostrophic equation.

**Answer:** Thank you for pointing out this issue. The comparison is removed.

**15 - 17) Figure 5:**

- Generally, observation error variances are much smaller than forecast error variances from free runs. The observation error variances used in this study are quite large, as they are dominated by the last 100 years with quite large forecast error variances in the free run. Since there are no model errors in the perfect twin experiments in this study, observation errors should be set much smaller than the current values. For reference, forecast errors approach around five over time in the Lorenz-63 and -96 models, while observation errors are prescribed to be one (i.e., 20% smaller).
- Please present temperature and stream function values using their respective units to enable comparison with practical systems
- A model timestep of 0.1, corresponding to 16 minutes, is likely to be too short. Model drift continues even after 200 years, and the error doubling time appears to be around 100 years, which is much longer than the error doubling time of 2 days in the atmospheric system (Lorenz, 1996). Please clarify how the authors define the timescale.

**Answer:** The observation error is chosen as 70% of the standard deviation of a long model trajectory. Therefore, as shown in the original Fig. 4, the observation errors are not spatially homogenous. This represents regionally high observation errors that is common in ocean particularly satellite observations. Moreover, even though it is common to use an observation of one for theoretical Lorenz systems, observation uncertainty can be high in real world. Further, the choice of the observation error does not affect the primary purpose of demonstrating an application of pyPDAF.

As discussed in De Cruz et al. (2016), all variables including temperature and streamfunctions are non-dimensionalised. Considering the idealised experimental setup, we deem it unnecessary to present dimensionalised variables and compare it with realistic systems.

The model parameters and time steps of the experiment follow Tondeur et al. (2020). In this study, as proposed in De Cruz et al. (2016), a 4th order Runge-Kutta scheme is used for numerical integration. The accuracy of numerical integration increases with

smaller time step. Increased model time steps can lead to higher numerical integration error which is not intrinsic to the model dynamics.

Based on Fig.4 of Tondeur et al. (2020), with our experiment setup of 36st, the largest Lyapunov exponent is  $0.1 \text{ day}^{-1}$ . This corresponds to an error doubling time of around 2 days. It is unclear to us how the reviewer reached the conclusion of an error doubling time of 100 years.

18) The minimum ensemble size is 10 in the LETKF-based Lorenz-96 system with 40 variables. However, the authors set the ensemble size to 16 members in a system with dimensions on the order of  $10^4$ – $10^6$ . Please demonstrate that an ensemble size of 16 members is sufficient for this system.

**Answer:** The required ensemble size of our experiment setup is tested in Tondeur et al. (2020). Our experiments use the same dynamical configuration. The only difference is that we assimilate in the grid point instead of spectral space. We explained the relation between the ensemble size and model dynamics in comment #12. In Sect.3.1, we have: "As shown by Tondeur et al. (2020), DA in the model configuration using 36 spectral coefficients can achieve sufficient accuracy with 15 ensemble members."

19) In ETKF and LETKF, the most computationally expensive part is applying eigen-decomposition to a square matrix of size equal to the ensemble size by ensemble size. This study intentionally reduces the ensemble size to 16, but operational systems typically require an ensemble size on the order of 102 to achieve sufficient accuracy. Therefore, it is necessary to assess the sensitivity to ensemble size using a high-dimensional system in order to investigate the computational performance of PDAF and pyPDAF.

**Answer:** The investigation of computational cost focuses on the differences between pyPDAF (Python) and PDAF (Fortran). In both cases, the eigen-decomposition is performed in Fortran-based PDAF routines. Therefore, the consideration of eigen-decomposition is not the reason for our choice of the ensemble size.

However, we agree that increasing the ensemble size can lead to more intensive numerical computations, and we acknowledge the need for investigate the sensitivity to ensemble size. Hence, we add a new experiments with an ensemble size of 64 and 128. This shows that an increased ensemble size reduces the relative overhead between Python and Fortran in pyPDAF as the numerical computations take a higher proportion of the total computational time compared to a small ensemble.

20) L333–336: If the authors intend to maintain a dynamical balance in the initial conditions, it would be better to extract them from the free run rather than applying second-order exact sampling. Multiplying by a factor smaller than one may degrade the dynamical balance.

**Answer:** We agree that utilising a free run can avoid any issues in dynamical consistency. However, part of the reason for using second-order exact sampling is to demonstrate the functionality of generating ensembles from a covariance matrix in PDAF. Also, generating a covariance from a long model trajectory can also provide more statistical information of the system than randomly selecting snapshots of the free run.

We add one sentence here, which hopefully clarify the issue: "PDAF provides functionality to generate the ensemble. Here, to demonstrate its functionality, we use second-order exact sampling (Pham, 2001), in which the ensemble is generated from a covariance matrix."

21) L326: Please clarify why a daily assimilation interval is chosen in this study, given that a 6-hour interval is typically used in the atmospheric data assimilation community.

**Answer:** We add: "This is in line with the experiment setup in Tondeur et al. (2020)."

22) L345 and elsewhere: The term "significant" should only be used if statistical tests are applied.

**Answer:** We removed all "significant".

23) Please clarify whether the authors calculate RMSEs and ensemble spread for forecasts or analyses throughout the manuscript (e.g., forecast RMSEs).

**Answer:** We always calculate the analysis RMSE and ensemble spread except for the free run in this paper as suggested by the descriptions and figure captions.

24) L366–367: Please clarify how the authors control error growth and forecast errors.

**Answer:** We add "controlled by DA" here.

25) L368–369: The explanation of analysis accuracy at grid points without observations is not reasonable, as the authors have not consider the impact of ensemble correlation.

**Answer:** We rephrased the sentences to: "This is due to the fact that the model fields are rather smooth leading to long ensemble correlations."

**26 - 27) Figure 6:**

• Please show not only the RMSEs but also the ensemble spread for comparison at the same time.

• Please apply a paired t-test to compare the RMSEs between weak and strong coupled data assimilation.

**Answer:** Following the suggestion by reviewer 3, to keep the CDA results discussion concise, we decide not to provide additional ensemble spread statistics here.

Also, we removed paragraphs for discussions on the SCDA and WCDA comparison to ensure they are not the focus of this study. Besides, a t-test here can provide little additional information for SCDA and WCDA in general as we only perform one specific MAOOAM configuration for demonstration purpose. Hence, we do not provide an additional t-test here.

28) The term "error" typically refers to an instantaneous error, whereas "RMSE" represents the statistical expectation of errors. Please use these terms appropriately to clarify the distinction between them.

Answer: Changed.

29) L389: Please clarify what is meant by "transient atmosphere processes."

**Answer:** The phrase is removed.

30) L390–394: Please demonstrate that the ocean exhibits a 60-year timescale in the stream function field, which results in minimum RMSEs at a 60-year smoothing window.

**Answer:** This is removed.

- 31-32) Subsection 5.2:
  - Figures 8 and 9 show computational times at the analysis timestep on a logarithmic scale, making it difficult to directly compare the differences between PDAF and pyPDAF. Please clarify these differences by providing the ratio.
  - Please include a description of the total computation time for one assimilation cycle, including both the forecast and analysis steps.

**Answer:** We acknowledge that it is difficult to understand the real computational time in logarithmic scale. However, it appears to be the only graphical solution due to the sheer differences in computational time.

To mitigate the limitation of the figures, we present computational time of selected experiments in Tab. 2, 3 and 4 with the ratio between the Python and Fortran implementations.

We do not see the need for including the computational time of forecast because they are implemented by the same Fortran code, leading to same forecast time. The forecast step is also irrelevant to the comparison between pyPDAF and PDAF.

33) L431: The term "very" is subjective. Please describe this in a more objective manner.

Answer: Removed.

34) L497: This sentence is inconsistent because the computational times for PyPDAF and PDAF are different. Please revise it to reflect the results obtained in this study.

Answer: We changed "the same" the "similar".

**Minor comment**

35) L5: "exists"  $\rightarrow$  "are"

**Answer:** Done

36) L5 and elsewhere: "need"  $\rightarrow$  "demands"

Answer: Done.

37) L15-16: Incorrect spelling of LEKTF (Local Ensemble Transform Kalman Filter).

**Answer:** Changed.

38) L19 and elsewhere: "weather and climate" → "atmosphere and ocean"

Answer: We believe that "weather and climate" is more inclusive than "atmosphere and ocean".

39) L29: Please spell out "DAPPER".

Answer: Done.

40) L38: Please specify which models are coupled.

Answer: Done.

41) L39 and elsewhere: There are no URLs, although the authors mention the date of last access.

Answer: The URLs are given as references. This follows the journal policy on webpage references (https://www.geoscientific-mode net/submission.html#references)

42) L74: "initialization"  $\rightarrow$  "initial"

**Answer:** "initialisation shock" is a term used by the cited reference Smith et al. (2015). This sentence is removed in the revised manuscript.

43) L78: Incorrect grammar.

Answer: Rephrased.

44) L97: "counter"  $\rightarrow$  "mitigate"

Answer: Changed.

45) L119: "ensemble-based 3DVar" → "ensemble variational data assimilation"

**Answer:** This sentence is removed.

46) L137 and elsewhere: "observation vectors and error covariance matrix" → "observations and observation error covariance matrix" Answer: We changed two occurrences of "observation vector". However, we keep the phrase when describing use of OMI in pyPDAF in Sect.2.4 because it should be treated as a vector there.

47) L139: Please specify the "direct" observation operator.

Answer: We change it to: "an observation operator for observations located on grid points"

48) L142: "would be"  $\rightarrow$  "is"

Answer: Changed.

49) L179: Please add the last access date.

Answer: Done.

50) L266: "coupled"  $\rightarrow$  "implemented", "implemented"  $\rightarrow$  "written", "that is coupled directly"  $\rightarrow$  "and is implemented"

Answer: Done.

51) When connecting two sentences, please insert a comma before "and" to enhance readability.

Answer: Done.

52) The use of "respectively" seems incorrect. A comma should be added before "respectively". For example, insert "each" before "Fortran" and remove "respectively" in L268.

**Answer:** In examples of the Cambridge dictionary, a comma only appears for American English version (https://dictionary.cambridge.org/dictionary/english/respectively, last accessed: 2025-03-25). Therefore, we don't think it is essential here.

53) Eqation (2): Please add an explanation for "n".

Answer: Added.

54) L315: Please specify the meaning of "ensure ... model state".

Answer: This is rephrased as "ensure that the initial state corrected by the DA follows the trajectory of the dynamical model"

55) L329: Please specify a forgetting factor. Is this the same as the relaxation parameter in the relaxation-to-prior perturbation and spread methods?

**Answer:** We add the following explanation: "The forgetting factor (Pham et al., 1998) is an efficient approach to multiplicative ensemble inflation in which the covariance matrix is inflated by the inverse of the forgetting factor as shown in the formulation in Nerger et al. (2012)"

56) L338: Please remove "generally".

**Answer:** Done

57) L357–358: Please check the meaning of the sentence.

Answer: Rephrased.

58) L464: "respectively" should be added after "12g state".

Answer: Done.

**Reviewer 2**

Thank the authors for this revision. My comments have been well addressed, and I have no further issues. The manuscript can be accepted.

**Answer:** We thank the comments and suggestions from the reviewer.

**Reviewer 3**

This manuscript introduces a newly developed Python interface to the Fortran-based Parallel Data Assimilation Framework (PDAF) software, pyPDAF. This tool aims to ease the development of new data assimilation systems by utilizing the Python programming language while only sacrificing the computational speed to an acceptable extent. It demonstrates an example of coupled data assimilation (CDA) using the Modular Arbitrary-Order Ocean-Atmosphere Model (MAOOAM) and ensemble transform Kalman filter (ETKF)/local ensemble transform Kalman filter (LETKF) algorithms, showing that the CDA systems developed based on PDAF and pyPDAF both work correctly and produce identical results, with their computational speeds measured for comparison. The idea of developing a Python interface to an existing data assimilation software package targeting efficient parallel computing is very useful to the community and worth publication. However, I find that this manuscript may need to be improved for its readability and strategy to present this topic. Therefore, I recommend that the manuscript can only be considered for publication after a major revision.

Answer: We thank for the reviewer's opinion that the package is useful to the community and worth publication.

**Major comments**

Overall, the strategy to present the topic of this manuscript may be reconsidered. The primary purpose of this article should be to introduce this newly developed pyPDAF, describe its concept, strengths, and weaknesses, and thus promote it to the community. A use-case example certainly needs to be included in this article, but it does not need to be too complicated or advanced in its experimental design or the presentation of the results. Several parts of the contents regarding the data assimilation methods and experiments may be shortened or removed. On the other hand, some other information about the development of data assimilation systems using pyPDAF should be better detailed or presented, such as the comparison of implementation difficulties between using PDAF and pyPDAF. I find that the manuscript has been improved in response to the comments of Referee #2 in the previous round of the review, with which I share similar opinions, but room for improvement still exists.

**Answer:** The DA methods described here are mainly existing methods in (py)PDAF. We have reformulated and shortened the description for better readability. The experimental description is mainly for the reproducibility purposes. We have shortened these descriptions in the revised manuscript.

We add a section "Comparison of pyPDAF and PDAF implementation of CDA" to compare the implementation difficulties between the Fortran and Python version using lines of code. We also provide the following description in the conclusion: "The advantage of pyPDAF in terms of the ease of implementation is reflected by a comparison of the number of lines of code by user-supplied functions in the SCDA setup. The pyPDAF implementation consistently uses fewer lines of code showcasing the requirement for a lower implementation effort than PDAF implementation."

1. I feel that the authors intended to provide much information and demonstrate some scientific findings regarding CDA in this manuscript; however, these may not be all necessary, considering that the focus of this manuscript should be introducing a new software tool for data assimilation development. Regarding the data assimilation experiment, I think its most important aspect should be to serve as an example of the data assimilation development using PDAF and pyPDAF, and the scientific insight may not be of the first concern. Therefore, I would suggest keeping the experimental design and the analysis and presentation of the results as simple as possible, so readers can easily understand the experiment results and focus on understanding the pyPDAF. Since the authors referred a lot to Tondeur et al. (2020), which used the same MAOOAM model, one option may be (only if the authors think this is appropriate) to repeat or mimic a few experiments in Tondeur et al. (2020) but using PDAF and pyPDAF so that the authors could save some words describing the experiments and interpreting the results.

In addition, a review of the ensemble-based data assimilation methods (Section 2) may not need to be too comprehensive as long as sufficient information relevant to this study is provided; for example, the particle filter method is not used in this study, so it may not be reviewed in too much detail. Besides, the ensemble generation method (i.e., "second-order exact sampling") is not very relevant to this study, either, as long as the approach is reasonable and a spin-up period is excluded from the analysis of the results (as in Lines 333-341).

**Answer:** In deed, the coupled data assimilation is not the focus of this study. We removed some descriptions of CDA in the abstract and the introduction. We also shortened the description of model and experiment setup in the revised manuscript.

Our experiment setup follows Tondeur et al. (2020) with the following differences:

- (a) DA is performed in spectral space in Tondeur et al. (2020) but we chose to perform DA in the physical space.
- (b) Observation errors are slightly higher in our experiment setup.

This is because our setup can be used to conduct experiments for different number of grid points with the same dynamical setup. Besides, using observations in actual, grid point, spaces is what is done in real DA applications.

We have moved the ensemble data assimilation section to be a subsection under the PDAF and pyPDAF section as Sect. 2.2. The subsection is repurposed to describe available DA methods in PDAF instead of a general introduction to ensemble DA.

2. The authors added Section 3.3 in the previous review process describing the things a developer needs to take care of from the aspect of several pyPDAF library interfaces, which was good. However, some of these contents appear too technical and too much like technical documentation of the software, but not easily understood by readers without having used the software. On the other hand, some critical information is still missing or not clearly presented: (1) to run the CDA experiments in the current study, what exactly are the programming tasks one needs to do by using PDAF and pyPDAF; how many "user-supplied functions" (simply listing them) are needed to be written by the users to fulfill the capability of running the current CDA experiments? (2) Were all the user-supplied functions written in Fortran and in Python, respectively, in the experiments using PDAF and pyPDAF? For example, for the spectral transformation calculation in Eqs. (1) and (2), were they coded separately in Fortran and Python in the two experiments? This information is important for readers to understand the relative implementation difficulties of using PDAF and pyPDAF.

**Answer:** We agree that Sect. 3.3 (now 2.4) is quite technical. To improve the readability, we have revised the section and put technical details into parenthesis. We hope this can reduce the intrusion of technical details in the main text while still providing a sufficient explanation of the user implementation with PDAF.

We add a section "Comparison of pyPDAF and PDAF implementation of CDA" to compare the implementation difficulties between the Fortran and Python version using the number of lines of code.

We add the following explanation for the implementation of Eqns. (1) and (2) which was mentioned in Sect. 4.2: "In this study, for the sake of efficiency, the transformation between spectral modes and grid points are implemented in Fortran. In pyPDAF systems, the Fortran transformation routines are used by Python with "f2py". This implementation ensures that the numerical computations do not render rounding errors when conducted in different programming languages. Moreover, it also demonstrates that the computation intensive component of user-supplied functions can be sped up by optimised Fortran code."

3. An important result I expect to see is the identicality of the data assimilation experiment results using PDAF and pyPDAF. The author did describe it but only in a brief sentence: "The online DA systems using PDAF and pyPDAF produce quantitatively the same results in all experiments up to machine precision." (Lines 353-354) I feel that this important aspect may deserve a bit more detailed discussion. In particular, given that a lot of user-supplied functions are written in different programming languages, it seems unlikely to me that their results can be "the same up to machine precision" It would be helpful if the authors could provide precise numbers of the analysis RMSEs of the two experiments using PDAF and pyPDAF.

**Answer:** This is indeed a relevant aspect. If the actual numerical computation is conducted in different languages, even though the differences of numerical computation is minimal in one time step, they can get magnified in nonlinear systems. In our experiments, the user-supplied functions are used to read observations and assign forecast fields to state vectors of Fortran PDAF arrays. User-supplied functions do not perform any numerical computations. Therefore, the user-supplied functions do not affect the DA step. However, our experiments indeed suggest divergent results if numerical computations are involved with two programming languages. For example, the transformation between spectral coefficients and grid point values. These motivate us to use the same Fortran subroutines and call them from Python in our experiments.

We add the following paragraph in the beginning of Sect. 4: "The online DA systems using PDAF and pyPDAF produce quantitatively the same results in all experiments up to machine precision. This is because, the user-supplied functions mainly perform file handling and variable assignments, but no numerical computations. An exception is only the spectral transformation described in Sect. 3.1. To ensure comparable numerical outcome, the numerical computations that affect the forecast and analysis, in particular the spectral transformation, are all conducted in Fortran in this work. These Fortran implementations are used by Python user-supplied functions using "f2py". Note that, when numerical computations involve different programming languages, the model trajectory of the nonlinear system could differ because of errors in the initial conditions arising from rounding errors."

4. The comparison of the computational performance of PDAF and pyPDAF is undoubtedly an important part of this study. The authors attempted to state that the computational speed of pyPDAF is only slightly slower than PDAF, especially when they are used with high-dimensional systems. However, from the results presented, it seems to me that their difference is actually not very small, particularly noting that Figs. 8 and 9 are plotted on a logarithmic scale, which may visually underestimate the differences. In addition, the study shows that in the case of LETKF (filters with domain localization), the difference in computational speed can be even more significant if the additional "PDAFlocal" module is not developed. Although this issue can be satisfactorily mitigated by the additional development presented, it also implies that the degree of the computational speed loss of using pyPDAF compared to PDAF could be very different case by case (different filters, observation operators, etc.). I feel that these results do not significantly detract from the value of pyPDAF, as enabling rapid development of data assimilation systems remains crucial. However, I suggest the authors moderate their claims about the advantage of pyPDAF in the computational aspect and clearly describe the limitations.

**Answer:** We understand the concern of using figures on log-scales. The reason for plotting on the log-scale is primarily due to the drastic differences of computational time between different spatial resolution and partially different functionalities instead of the differences between the pyPDAF and PDAF implementation.

To mitigate this issue, we add "in log-scale" in the y-axis of the figures and make the caption of "log-scale" bold. We also added Tab. 2, 3 and 4 to present computational time of selected experiments. In these tables, we include the ratio of computational time between pyPDAF and PDAF.

To address the concerns on the computational efficiency, we specify that the 13% slow down of LETKF is a result of our specific example. We also state that the overhead of the computational cost can vary case by case. For example, we changed the last sentence in the abstract to "The study also shows that pyPDAF can be used with high-dimensional systems with little slow-down per analysis step of only up to 13% for the localized ensemble Kalman filter LETKF in the example used in this study. The study also shows that, compared to PDAF, the overhead of pyPDAF is comparatively smaller when computationally intensive components dominate the DA system. This can be the case for systems with high-dimensional state vectors."

We also add the following discussions in the manuscript: "We recognise that the exact computational time can be case-specific. For example, we can postulate that, compared to this study, the overhead can be comparatively smaller for computationally intensive user-supplied functions where JIT can be used. This could be the case when correlated observation error covariances are used. Even though this study only investigates the commonly used ETKF and LETKF, the relative run times of pure PDAF and pyPDAF should be similar for other global and local filters. This expectation results from the algorithmic similarity of many filters and the fact that the user routines which are coded in Python when using pyPDAF are mainly the same. However, the overhead may also vary depending on the DA algorithms, in particular for variants of 3DVar."

We stated that the "PDAFLocal" module is designed specifically to mitigate the overhead in pyPDAF in Sect. 4.2. "To overcome the specific run time issue of 'g2l state' and 'l2g state', we developed a *PDAFlocal* module in PDAF, included in release version 2.3..." In the conclusion, we also recognise the possibility to improve the efficiency in future development: "We recognise that the computational cost of the pyPDAF and PDAF can vary case-by-case. Our results demonstrate that the additional "PDAFlocal" module was essential to mitigate the computational overhead in the case of domain localisation. When pyPDAF is used for other DA algorithms and applications, potential efficiency gain can be implemented in future releases of both PDAF and pyPDAF as both pyPDAF are still under active development and maintenance."

Please note, the "PDAFlocal" is now used as the default, so that the overhead occurring when not using "PDAFlocal" is no longer a concern. The discussion on the overhead without "PDAFlocal" and the solution obtained by introducing it is now intended to serve as a example of issues one can encounter when combining Python code with Fortran.

**Minor comments:**

1. Lines 14 (in Abstract) and 34: These are the first appearances of the terminology "user-supplied function" in PDAF. In my understanding (after reading more contents in the manuscript and the PDAF documentation), it stands for the additional code users need to write to complete a data assimilation system based on PDAF, but this is not very straightforwardly understood in the beginning. I suggest that this term be better explained in its first appearance.

**Answer:** In the abstract, we avoid the word "user-supplied function" and use a more descriptive sentence: "This study demonstrates that pyPDAF allows for PDAF functionalities from Python where users can utilise Python functions to handle case-specific information from observations and numerical model."

In the introduction, we provide an explanation of the user-supplied functions: "In this generic framework, DA methods accommodate case-specific information about the DA system through functions provided by users including the model fields, treatment of observations, and localisation. These functions are referred to as *user-supplied functions*."

2. Section 4.2, experiment design: What is the length of the cycled data assimilation experiments? Is it 300 years? This seems to be implied in Fig. 5 but is not explicitly provided.

**Answer:** We add a sentence: "The DA experiments are then run for another  $9 \times 10^5$  time steps which is around 277 years."

3. Lines 329-330: What does the "forgetting factor" mean? Does it represent some parameters in a specific covariance inflation scheme?

**Answer:** The 'forgetting factor' relates to a computationally particularly efficient scheme for multiplicative inflation introduced by Pham et al. (1998). While the factor itself specifies the inflation, it is also synonymous for the method. Explaining details would be beyond the scope of the manuscript. To this end, we only add the following explanation: "The forgetting factor (Pham et al., 1998) is an efficient approach to multiplicative ensemble inflation in which the covariance matrix is inflated by the inverse of the forgetting factor as shown in the formulation in Nerger et al. (2012)."

4. Lines 384-385, "the time-averaged RMSE of fields that are smoothed in time by a moving average as a function of the averaging time-window": I find that the meaning of Fig. 7 is difficult to understand. Does it mean first applying a moving average (with variable time-window lengths in the x-axis) to the spatial RMSEs across the 300-year experiment period (related: Minor comment

**2) and then computing the temporal average of the moving average results? If this is correct, the scientific meaning behind this figure remains difficult to me: Why do the authors want to do the "double temporal average" (average of moving average)? Is this meaningful? Following my Major comment #1, to keep the experiment results as simple as possible, this figure may be removed if it is not critical to the theme of this manuscript.**

**Answer:** The moving average in time is applied to the actual model field to obtain a temporally smoothed field, and the RMSE is calculated for the smoothed field afterwards. The idea was to compare the RMSE of slow processes of the system, e.g., seasonal climate. We removed the figure and corresponding discussion.

5. Lines 402-403: Why is the data assimilation calculation performed on a single processor instead of 16 processors used for running ensemble model forecasts? Is there any practical restriction of PDAF to parallelize the data assimilation calculation with an arbitrary number of processors?

Answer: In our experiments, we tested both ETKF and the local ETKF (LETKF). ETKF can be easily parallelised with a distributed computation of the global transform matrix. The LETKF can be embarrassingly parallel for each local domain. The default MPI parallelisation strategy in PDAF utilises the domain decomposition of numerical models. For example, assume we have 3 ensemble members, each of which is decomposed to 4 processors, during the forecast, PDAF collects the full ensemble from the 3 ensemble members to the first ensemble member without affecting the domain decomposition. During the analysis step, the domain-decomposed ensemble states are processed on the 4 processors of the first ensemble member. The LETKF is performed on local domains on these processors. This approach was found to be efficient in particular when using models with unstructured grids which cannot easily be further decomposed. However, an analysis using more processors is possible with a change in PDAF's communication routine. In our experiment, the numerical model is not parallelised leading to a single processor for LETKF. In PDAF, one can further utilise shared memory parallelisation with OpenMP. However, this is less straightforward in pyPDAF because the global interpretor lock (GIL). To enable shared memory parallelism, frequent acquirement and release of the GIL is needed which can degrade the performance of LETKF.

We add the following paragraph: "The ETKF has a straightforward parallelisation since the global transform matrix can be computed in a distributed form followed by a global sum. The LETKF is embarrassingly parallel for each local domain after communicating the necessary observations. Each processor can perform LETKF independently for their local domains. In PDAF, the parallelisation of both ETKF and LETKF is implemented in combination with domain decomposition of the numerical model. In this study, no domain decomposition is carried out for the numerical model itself. Thus, all local domains are located in one single processor for LETKF. The parallelisation strategy of PDAF is further explained in Nerger et al. (2005) and a pyPDAF documentation is available (Parallelisation Strategy, Accessed: 20 March 2025)."

6. Figure 8, Lines 423-424: The "MPI" communication time is long and accounts for a large portion of the total computation time (in both PDAF and pyPDAF). Given that the number of processors (16?) is not many, why does the MPI communication time take so long? Could the authors briefly explain where this MPI communication time is mostly spent?

**Answer:** In this study, because DA is performed serially, the MPI communication only occurs when the ensemble is collected from the state vector in each ensemble member before the analysis and distributed to the state vector after the analysis. This involves data exchange between different processors. Therefore, we see an increased MPI communication time with larger state vector. The lack of computer memory can also lead to slow MPI communications due to the use of swap memory. In the revised manuscript, we increased the memory allocation for each experiment that leads to reduced MPI communication time.

We add the following sentence: "In this study, the MPI communications are only used to gather an ensemble matrix from the state vector of each ensemble member located at their specific processor. These communications are internal to PDAF, and are not exposed to users, which show little differences between pyPDAF and PDAF system."

7. Lines 444-445: What exactly is the localization length scale or cut-off radius used in this study? What do the authors mean by the "1 spatial unit"?

**Answer:** We clarified this by writing: "Here, we choose a domain with  $257 \times 257$  grid points to assess the LETKF with a cut-off localisation radius of 1 non-dimensionalised spatial unit. This corresponds to 3000 km covering around a third of the domain."

8. Figure 9: Why are there two missing bars in the "no. domains" part? Are they excessively small so it does not appear in this figure? This needs to be corrected or explained.

Answer: Yes. The computational time is below  $10^{-5}$  seconds in Fortran as they are a simply a constant assignment which may even be inline optimised by the compiler. We add the following explanation in the manuscript: "The 'no. domains' user-supplied function takes  $\sim 1.8 \times 10^{-5}$  s per analysis step for pyPDAF system but only  $\sim 1. \times 10^{-6}$  s is taken by the PDAF system. The latter can be negligible when every 8 grid points are observed. In this user-supplied function, only one assignment is executed in the user-supplied function. Therefore, the overhead is primarily a result of conversion between the interoperation between Fortran and Python. This operation has little impact on the overall efficiency of the system."

We also add a note in the figure caption: "The computational time of PDAF system for 'no. domains' is negligible when every 8 grid points are observed which lead to an empty bar."

**References**

- Anderson, J. L.: An Ensemble Adjustment Kalman Filter for Data Assimilation, Monthly Weather Review, 129, 2884 2903, https://doi.org/10.1175/1520-0493(2001)129\(2884:AEAKFF\)\(2.0.CO;2, 2001.
- Bocquet, M.: Ensemble Kalman filtering without the intrinsic need for inflation, Nonlinear Processes in Geophysics, 18, 735–750, https://doi.org/10.5194/npg-18-735-2011, 2011.
- Bocquet, M., Gurumoorthy, K. S., Apte, A., Carrassi, A., Grudzien, C., and Jones, C. K. R. T.: Degenerate Kalman Filter Error Covariances and Their Convergence onto the Unstable Subspace, SIAM/ASA Journal on Uncertainty Quantification, 5, 304–333, https://doi.org/10.1137/16M1068712, 2017.
- De Cruz, L., Demaeyer, J., and Vannitsem, S.: The Modular Arbitrary-Order Ocean-Atmosphere Model: MAOOAM v1.0, Geoscientific Model Development, 9, 2793–2808, https://doi.org/10.5194/gmd-9-2793-2016, 2016.
- Nerger, L., Hiller, W., and Schröter, J.: PDAF The parallel data assimilation framework: experiences with Kalman filtering, in: Use of High Performance Computing in Meteorology, pp. 63–83, https://doi.org/10.1142/9789812701831\_0006, 2005.
- Nerger, L., Janjić, T., Schröter, J., and Hiller, W.: A Unification of Ensemble Square Root Kalman Filters, Monthly Weather Review, 140, 2335 2345, https://doi.org/10.1175/MWR-D-11-00102.1, 2012.
- Parallelisation Strategy: https://yumengch.github.io/pyPDAF/parallel.html, Accessed: 20 March 2025.
- Pham, D. T.: Stochastic Methods for Sequential Data Assimilation in Strongly Nonlinear Systems, Monthly Weather Review, 129, 1194 1207, https://doi.org/10.1175/1520-0493(2001)129/1194:SMFSDA\2.0.CO;2, 2001.
- Pham, D. T., Verron, J., and Christine Roubaud, M.: A singular evolutive extended Kalman filter for data assimilation in oceanography, Journal of Marine Systems, 16, 323–340, https://doi.org/10.1016/S0924-7963(97)00109-7, 1998.
- Simon, E. and Bertino, L.: Gaussian anamorphosis extension of the DEnKF for combined state parameter estimation: Application to a 1D ocean ecosystem model, Journal of Marine Systems, 89, 1–18, https://doi.org/https://doi.org/10.1016/j.jmarsys.2011.07.007, 2012.
- Smith, P. J., Fowler, A. M., and Lawless, A. S.: Exploring strategies for coupled 4D-Var data assimilation using an idealised atmosphere–ocean model, Tellus A: Dynamic Meteorology and Oceanography, https://doi.org/10.3402/tellusa.v67.27025, 2015.
- Tondeur, M., Carrassi, A., Vannitsem, S., and Bocquet, M.: On temporal scale separation in coupled data assimilation with the ensemble kalman filter, Journal of Statistical Physics, 179, 1161–1185, https://doi.org/10.1007/s10955-020-02525-z, 2020.
- Whitaker, J. S. and Hamill, T. M.: Ensemble Data Assimilation without Perturbed Observations, Mon. Wea. Rev., 130, 1913–1927, 2002.
- Yu, H.-C., Zhang, Y. J., Nerger, L., Lemmen, C., Yu, J. C., Chou, T.-Y., Chu, C.-H., and Terng, C.-T.: Development of a flexible data assimilation system for a 3D unstructured-grid ocean model under Earth System Modeling Framework, Ocean Modelling, 196, 102 546, https://doi.org/https://doi.org/10.1016/j.ocemod.2025.102546, 2025.

---

## Author Response (AR3)

**Reviewer 1**

**General comment**

The authors have satisfactorily addressed all my previous comments. I only have a few minor suggestions and comments listed below. I recommend that the manuscript be accepted after resolving these minor points.

**Minor comments:**

1. Lines 121-123: Suggest rephrasing the sentence as "Other typical filtering algorithms, such as ensemble adjustment Kalman filter (EAKF, Anderson, 2001) and ensemble square root filters (EnSRF, Whitaker and Hamill, 2002), are not implemented in current releases but are planned to be included in future releases."

**Answer:** We rephrase the sentence based on the reviewer's suggestion with a slight adjustment. The last part of the sentence is written in a clearer fashion due to newer development in PDAF.

"not implemented in PDAF V2.3 used in this work, but were introduced in the newer release V3.0."

2. Line 232: "with the OMI functionality, only three user-supplied functions need to be implemented." It is not clear from the following text which three user-supplied functions are needed.

**Answer:** Thank you for the point. We rephrase the sentences and add the following content in Sect.2.4:

To handle different observations, with the OMI functionality, only three user-supplied functions need to be implemented:

**Func 1:** a function that provides observation information like observation values, errors and coordinates  $(dim\_obs = init\_dim\_obs(step, dim\_obs))$  where its primarily purpose is to provide the dimension of observation vector  $(dim\_obs)$  to PDAF,

**Func 2:** a function that provides the observation operator  $(m\_state\_p = obs\_op(step, dim\_p, dim\_obs, state\_p, m\_state\_p))$ , where the observation operator transforms the state vector  $(state\_p)$  into observation space  $(m\_state\_p)$ ,

Func 3: a function that specifies the number of observations being assimilated in each local domain  $(dim\_obs\_l = init\_dim\_obs\_l = (domain\_p, step, dim\_obs, dim\_obs\_l)$ ).

3. Lines 386-395 and 401-404: "In WCDA, each model component performs DA independently ... and define the local state vector as either the atmosphere or ocean variables." and "Compared to the WCDA, ... and does not require special treatment." These sentences do not fit well in the context of Section 4.1 "Skill of data assimilation." They may be more appropriately placed in Section 3.2 "Experiment design" or somewhere else.

**Answer:** They are moved to Sect. 3.2 "Experiment design".

4. Line 437: "the ratio of total computational time" should be "the ratio of computational time for 'pre-post'."

Answer: Done.

5. Line 441: "... but the ratio is only 2.04 and 3.58 for 'distribute state' and 'collect state' ..." The ratio should be 3.58 and 8.60 according to Table 2. However, this correction makes the reduction of pyPDAF overhead much less impressive, so the sentence may need to be further revised.

**Answer:** We corrected the mistake and slightly changed the wording:

The overhead in the pyPDAF system is less affected by the dimension of the state vector for the distribution and collection of state vector (labelled 'distribute state' and 'collect state') because these functions only exchange information between model and PDAF without intensive computation. For example, the pyPDAF system takes 3.82 and 8.89 times of computational time of the PDAF system for 'distribute state' and 'collect state' respectively on a  $129 \times 129$  grid, which is similar to the ratio of 3.58 and 8.60 on a  $2049 \times 2049$  grid.

6. Table 3: Please clarify why the wall clock times of all components do not sum to the "total" time shown in the last row of the table.

**Answer:** To match the total time, we added in Tab. 3 the omitted 'local obs. search' entry which does not affect our main results. We also added:

The similar computational time applies for the case when PDAF search for local observations for analysis local domains due to its intensive numerical computation.

7. Line 562: I am not sure where the number "70%" comes from.

**Answer:** This is a mistake from revision. We rephrase the sentence to:

In the scope of our specific experiment setup, compared with PDAF, our benchmark shows that, depending on the size of the state vector and ensemble, from 28% to around three times more time (see Tab. 2 and 4) is used by pyPDAF with the global filter while only 6% - 13% more time is required with a domain-localized filter when applying the Python DA system build with pyPDAF in a high-dimensional dynamical system.